# Mobilization of cholesterol induces the transition from quiescence to growth in *Caenorhabditis elegans* through steroid hormone and mTOR signaling

Kathrin Schmeisser [1✉], Damla Kaptan[1], Bharath Kumar Raghuraman[1], Andrej Shevchenko [1], Jonathan Rodenfels[1,2], Sider Penkov[1,3] & Teymuras V. Kurzchalia [1✉]

Recovery from the quiescent developmental stage called dauer is an essential process in *C. elegans* and provides an excellent model to understand how metabolic transitions contribute to developmental plasticity. Here we show that cholesterol bound to the small secreted proteins SCL-12 or SCL-13 is sequestered in the gut lumen during the dauer state. Upon recovery from dauer, bound cholesterol undergoes endocytosis into lysosomes of intestinal cells, where SCL-12 and SCL-13 are degraded and cholesterol is released. Free cholesterol activates mTORC1 and is used for the production of dafachronic acids. This leads to promotion of protein synthesis and growth, and a metabolic switch at the transcriptional level. Thus, mobilization of sequestered cholesterol stores is the key event for transition from quiescence to growth, and cholesterol is the major signaling molecule in this process.

[1] Max Planck Institute of Molecular Cell Biology and Genetics, Dresden, Germany. [2] Physics of Life (PoL), Technical University Dresden, Dresden, Germany. [3] Faculty of Medicine, Technical University Dresden, Dresden, Germany. ✉email: kathrin.schmeisser@mpi-cbg.de; t.kurzchalia@gmail.com

Environmental conditions such as temperature, population density, or the presence of competitor species, amongst many others, can impact the development of an organism at different levels. The most potent exogenous factor is the availability of nutrients and micronutrients, which have a dramatic effect on the growth of almost all species. However, lack of essential nutrients and other environmental factors are common and have led to multiple adaptation strategies, including the existence of stages of quiescence or other alternative developmental programs. This so-called developmental plasticity ensures the survival of an organism during harsh times by adjusting growth and physiology until conditions improve[1].

*C. elegans* is an excellent model to study developmental plasticity as it has evolved several strategies to adapt to stressful environmental conditions during its development. The free-living nematode relies mostly on microorganisms in organic material as a food source in the soil, its natural habitat in the wild[2]. However, at times of adverse environmental conditions, their development can be modulated or completely arrested in three different quiescent stages: the L1 diapause[3], the adult reproductive diapause[4], and the dauer larva state[5]. The latter is an alternative developmental stage. Late-stage L1 larvae can change their developmental trajectory and postpone reproduction by going through a preparatory second larval stage (L2d) and a quiescent dauer larval stage. This increases the chance of survival until environmental conditions improve and progeny can be sustained.

Several signaling mechanisms, including the insulin/IGF-1, TGF-beta, cyclic GMP, and steroid-signaling pathways control the ensuing changes in gene expression that lead to the characteristic morphological and metabolic changes typical of dauer development. These morphological changes include a modified cuticle, reduced radial diameter, and a sealed pharynx. All these are adaptions to harsh environmental conditions. During such conditions, their metabolism is substantially different compared to that of reproductive larvae. Indeed, they do not feed and rely on a catabolic mode of metabolism, using stored energy reserves to sustain essential physiological processes. To conserve these limited reserves, dauers are hypometabolic, which implies that heat production, aerobic respiration, and TCA cycle activity are significantly reduced[6]. At the same time, the glyoxylate shunt and gluconeogenesis are used for glucose generation from stored lipids[7,8]. Dauers can survive in this state for weeks to months, largely surpassing their life expectancy in reproductive mode.

Exit from the dauer state offers an excellent model system to study the transition from quiescence to growth. Two major processes should be among the events implicated in the transition: transcriptional reprogramming and return to full-scale protein synthesis. The transcription factors that determine the developmental states on the gene expression level in dauer or reproductive state have been studied extensively[9–11]. These include the nuclear hormone receptor DAF-12 and the FoxO member DAF-16. Whereas DAF-16 is mainly affected by insulin-like signaling via the insulin receptor orthologue DAF-2, DAF-12 is controlled by cholesterol derivatives called dafachronic acids (DAs). These are bile acid-like steroid hormones that occur in four isomers: two regioisomers, Δ4- and Δ7-DAs, which exist as 25R- and 25S-diastereomers. Their major biosynthetic step is performed by DAF-9, a cytochrome P450, hydroxylating cholesterol. In its DA-bound state, DAF-12 promotes reproductive development and suppresses dauer formation or facilitates dauer exit. When DA is not bound to DAF-12, dauer formation is induced[12,13]. This, however, requires very low or even absent DA levels in the cellular environment, because DA acts in extremely low concentrations, probably in the picomolar range. It is still not fully understood how the complete absence of DA can be achieved. One way of the reduction of DA might be the activity of the sterol methylase STRM-1. It has been shown that this enzyme reduces the amount of DA by modifying cholesterol in a manner such that DAF-9 cannot be used as the substrate to generate DA. However, only up to 50% of the sterol pool can be methylated[14]. As *C. elegans* cannot synthesize steroids de novo and must obtain them from their diet, another way to lower DA levels could be to reduce cholesterol digestion or to compartmentalize/sequester cholesterol within the animal. Such a compartmentalization mechanism has not yet been described though.

As mentioned above, the initiation of full-scale translation should be another crucial step in dauer exit. A known candidate for regulating this process is mTOR, a highly conserved serine/threonine protein kinase and part of the catalytic subunit of two protein complexes, mTORC1 and mTORC2[15]. Whereas both complexes can sense environmental conditions and regulate different cellular processes, mTORC1 is responsible for protein synthesis and growth. The major components of mTORC1 are the TOR kinase and the regulatory-associated protein of mTOR (RAPTOR). LET-363 and DAF-15 are the *C. elegans* orthologues, respectively[16]. It has been shown that both LET-363 and DAF-15 are required during development to progress through the larval stages[17]. Interestingly, knockout of *daf-15*/RAPTOR causes larval arrest at the L3 stage, which, in terms of developmental timing, coincides with the dauer diapause that is regulated by the steroid hormone pathway[18]. Hence, their shared sensitivity to cholesterol levels and developmental timing factors suggests the possibility that there is a cross-talk between mTORC1 and the steroid hormone pathway. However, such a cross-talk has not yet been investigated.

We decided to study dauer recovery using an unbiased approach by investigating proteomic changes that occur during the very early phase of dauer exit. Surprisingly, the key change observed was the depletion of sterol-binding proteins SCL-12 and SCL-13. It appears that this phenomenon is associated with the mobilization of cholesterol from internal pools. When *C. elegans* enters dauer, cholesterol is bound to SCL-12 and SCL-13 and is transported into the intestinal lumen and stored there throughout the duration of the dauer stage. However, upon dauer exit, the complexes of cholesterol bound to SCL-12 and SCL-13 are transported to the lysosomes, where the proteins are presumably degraded and cholesterol is released. Free cholesterol serves as the precursor of DA, which binds DAF-12, and subsequently causes a switch of developmental programs at the transcriptional level. Most importantly, cholesterol in lysosomes activates mTOR and in this way boosts protein synthesis and consequently growth. Our data show that the release of cholesterol from these SCL-12- and SCL-13-dependent internal deposits is the key event in the transition between dauer state and reproductive development.

## Results

**Degradation of SCL-12 and SCL-13 is a major proteomic change during the early phase of dauer exit.** To investigate early proteomic changes during the transition between the quiescent dauer larva and the reproductive stage, we performed a 2D-differential gel electrophoresis (2D-DIGE). We compared the protein landscape of wild-type dauers obtained from over-crowded medium before (labeled red) and during exit from the dauer state induced by exposure to food at low population density (labeled green). In contrast to our assumption that many new protein bands would appear during dauer exit due to the synthesis of proteins required for reproductive growth (in green), the most striking difference was that two abundant proteins, SCL-12 and SCL-13, completely disappeared after the first 4 h of dauer exit (two red spots in Fig. 1a; marked by arrows). SCL-12 and SCL-13 had been identified before as dauer-specific[19,20].

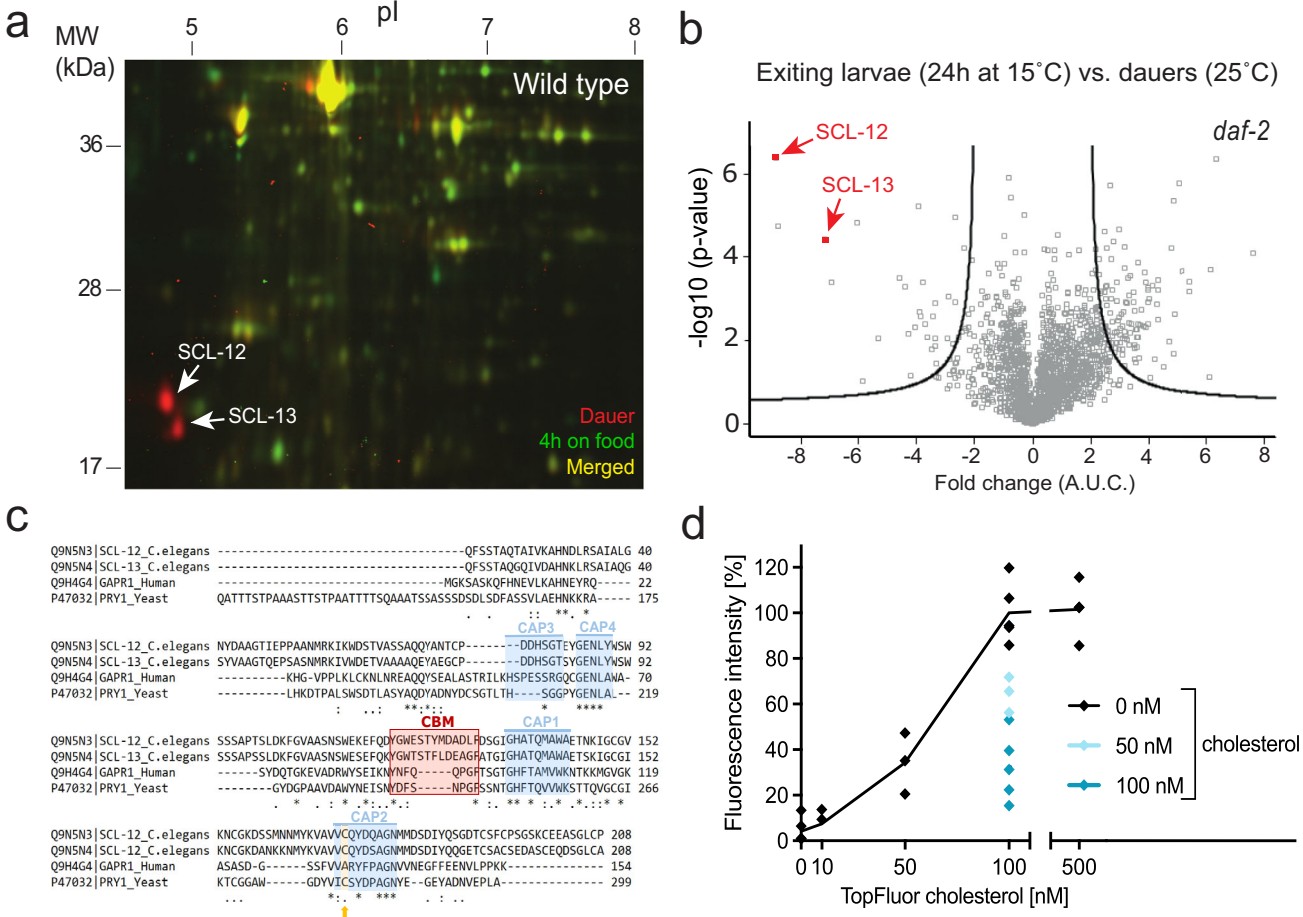

**Fig. 1 Degradation of SCL-12 and SCL-13 cholesterol-binding proteins is a major proteomic change during the early phase of dauer exit. a** Overlay of false-colored 2D-DIGE images comparing the wildtype (wt) starvation dauer proteomes before (red) and 4 h after (green) introduction of food. Yellow spots represent proteins with no major changes in concentration. The most striking changes were SCL-12 and SCL-13, which are depleted within 4 h on food. **b** The proteome of 24 h exiting dauer *daf-2(e1370)* animals compared with the dauer proteome. Fold changes indicate upregulated (>0) and downregulated (<0) proteins in exiting dauer larvae. Black lines designate significance, i.e. *p*-values < 0.05. Raw data are available via ProteomeXchange with the identifier PXD048087. **c** Alignment of *C. elegans* SCL-12 (UniProt ID: Q9N5N3_CAEEL), SCL-13 (Q9N5N4_CAEEL), human GLIPR2/GAPR1 (Q9H4G4) and yeast PRY1 (P47032) protein sequences. The functional signatures 1–4 of the CAP domain are highlighted in blue. Essential for cholesterol binding is the caveolin-binding motif (CBM; red) and a cysteine residue within the second signature of the CAP domain (orange). **d** Fluorescent intensity of purified SCL-12 protein that was incubated with various concentrations of TopFluor cholesterol (black) and pre-incubated with 50 nM (light turquoise) or 100 nM (turquoise) cholesterol for 1 h, bound to sepharose beads. Means and individual values of two experiments with at least two technical replicates; paired two-tailed *t*-test of 0 nM versus 50 nM cholesterol showed *p* = 0.0022 and 0 nM versus 100 nM cholesterol *p* = 0.001.

To obtain a more accurate picture of this potential proteomic switch, we performed label-free LC–MS/MS quantification of proteins in a strain that harbors the *daf-2(e1370)* allele. These mutant animals form dauers at a restrictive temperature of 25 °C and exit from the dauer state when shifted to the permissive temperature of 15 °C, providing an efficient genetic means to switch between metabolic states[21]. It must be noted that the exit of *daf-2* takes longer than in wild-type larvae[22]. Compared to the initial dauer state, exiting dauers 24 h after the switch to 15 °C displayed 142 proteins whose differential expression is significant. (Fig. 1b). Of these, 110 were up- and 32 were downregulated. Among the proteins with more prominent upregulation were a chromatin remodeller (GFI-1), a ribosomal protein involved in translation (RPL-28), and S-Adenosyl methionine synthetase (SAMS-1). SCL-12 and SCL-13 were among the most strongly decreased proteins in concentration (Fig. 1b), confirming the 2D-DIGE result in Fig. 1a.

**SCL-12 and SCL-13 are conserved cholesterol-binding proteins**. SCL(SCP-like)-12 and SCL-13 belong to a large family of proteins in *C. elegans* that display strong homology to the eukaryotic SCP/TAPS (Sperm-coating protein/Tpx/antigen 5/pathogenesis related-1/Sc7), or CAP (cysteine-rich secretory protein/antigen 5/pathogenesis related-1) superfamily. SCL-12 and SCL-13 align especially closely to the yeast CAP protein Pry1 (Fig. 1c). Pry1 binds lipids, among them cholesterol[23,24], and is implicated in the export of cholesterol acetate from the cytosol as part of a proposed lipid detoxification mechanism in yeast[25]. Other CAP proteins from arthropods and humans also bind cholesterol/lipids[26,27]. The caveolin-binding motif (CBM) and a conserved cysteine residue of Pry1 are responsible for free cholesterol and cholesteryl acetate-binding[23,24]. Within the CBM, the aromatic amino acids are essential to bind cholesterol[23]. SCL-12 and SCL-13 also possess a CBM with its corresponding aromatic residues, and a conserved cysteine in signature 2 of the CAP domain (C171; Fig. 1c).

We tested whether *C. elegans* SCL-12 binds sterol by performing an in vitro binding assay[23]. SCL-12 and SCL-13 are highly homologous (76.8% identity) and share the same promoter. Therefore, we tested the sterol-binding of SCL-12 alone as representative of the two. SCL-12 was expressed and purified from insect cells and incubated with a fluorescent analog of cholesterol (TopFluor cholesterol). As previously shown, this reagent behaves similarly to native cholesterol regarding intracellular transport or its incorporation into membranes[28]. Binding of TopFluor cholesterol to SCL-12 is concentration-dependent (Fig. 1d). Specificity of binding to cholesterol was further confirmed using a competitive approach where SCL-12 was pre-incubated with unmodified cholesterol, which decreased the binding of TopFluor cholesterol significantly (Fig. 1d). Thus, SCL-12 (and very likely SCL-13) are conserved sterol-binding proteins in *C. elegans* that bind cholesterol and perhaps other related sterols.

**SCL-12 accumulates in the dauer gut lumen and is transported back into the intestinal cells to be degraded during dauer exit.** To study where SCL-12 and SCL-13 exert their function, we generated a protein reporter in which SCL-12 is tagged with a mScarlet red fluorescent protein at its C-terminus and expressed it in the dauer-constitutive strain *daf-2* (SCL-12::mScarlet;*daf-2*; Fig. S1a). Expression of the protein starts in the intestinal cells in late L2d worms (i.e. the specific L2 larval stage that precedes the dauer larva), about 44 h after hatching at 25 °C (Fig. S1b). During dauer formation, fluorescence in the gut lumen increases (Fig. S1c). Afterward, SCL-12 is exclusively found within the intestinal lumen in fully formed dauers (Fig. 2a).

When exit was induced by a temperature switch in SCL-12::mScarlet;*daf-2* animals, the fluorescence in the gut lumen decreased dramatically and almost disappeared within 24 h (Fig. 2b; quantification in Fig. 2c). In parallel, faint fluorescence could be detected in the cytosol of the intestinal cells in punctate structures (Fig. 2b; 12 and 24 h time points). This fluorescence, however, also disappeared completely after 48 h. The same can be observed in SCL-12::mScarlet dauers in a wt background, where dauer exit was induced by the introduction of food, although the process occurs much faster (Fig. S1d, S1e).

**SCL-12 and SCL-13 sequester cholesterol in the gut lumen of dauers.** As SCL-12 binds cholesterol in vitro and SCL-13 is assumed to behave in the same way, we wondered whether they might also play a role in the transport and distribution of sterols in vivo. To visually follow the transport of cholesterol, we used the TopFluor cholesterol analog, which was also used in the in vitro sterol-binding assay. SCL-12::mScarlet-expressing *daf-2* animals were allowed to develop into dauer on plates in which cholesterol was replaced by TopFluor cholesterol. Notably, like SCL-12, most of TopFluor cholesterol accumulates in the lumen of the dauer gut (Fig. 2d, 0 h). Moreover, it follows the SCL-12::mScarlet fluorescence during dauer exit into punctate structures and becomes less visible in the gut lumen (Fig. 2d, 24 h). This is consistent with the idea that SCL-12 indeed binds cholesterol in worms that enter the dauer state and that SCL-12 sequesters cholesterol in the gut lumen during dauer, to release cholesterol and allow its degradation during dauer exit.

In order to clarify the role of SCL-12 and SCL-13 in the sequestration of cholesterol, we created a double-knockout strain introducing a 2.700 bp-deletion, spanning both the *scl-12* and *scl-13* genes on chromosome V (Fig. S1a). *scl-12&13* mutants were crossed with *daf-2* animals, hereafter referred to as *scl-12&13;daf-2*. Like *daf-2*, this strain produces 100% dauers at the restrictive temperature (25 °C), which, although being slightly smaller in size than *daf-2* alone (Fig. S2a), does not display any visible phenotype. *scl-12&13;daf-2* mutants survive 1% SDS treatment, and have the same oxygen consumption rate (OCR; Fig. S2b; 0 h time point) and survival as the parental *daf-2* strain (fig. S2c). However, the distribution of TopFluor cholesterol in *scl-12&13;daf-2* dauers significantly differs from that of *daf-2* control animals in that the gut lumen fluorescence is much lower (Fig. 2e; quantification in Fig. 2f). We also used a strain that harbors a knock-out of *scl-12* alone in a *daf-2* background, *scl-12;daf-2*. Here, we found no significant difference in TopFluor cholesterol in the gut, but much more variability between individual worms (Fig. 2f). This suggests that in most animals, SCL-13 can compensate for the loss of SCL-12, while in some animals this is not the case.

In summary, we conclude that SCL-12 and SCL-13 act redundantly to bind cholesterol and are pivotal for cholesterol sequestration and storage in the gut lumen in dauer larvae.

**Sequestration of cholesterol is necessary for appropriate dauer exit.** To investigate whether and how cholesterol sequestration influences dauer exit, we used *scl-12&13;daf-2* double and *scl-12;daf-2* single mutants. When we induced dauer exit by temperature switch to the permissive 15 °C, the double mutants only partially recovered, with about 40% of all animals remaining arrested (Fig. 3a). This partial phenotype could be explained by the existence of 25 other SCL genes in the genome of *C. elegans* that could also be involved in cholesterol sequestration. In worms that do eventually exit the dauer state, growth is significantly delayed (Fig. 3b) and they reach fertility two days later than the control (Fig. S2d). In *scl-12;daf-2*, only a small delay in dauer exit could be observed (Fig. S2e), again suggesting that SCL-13, together with the other unexplored SCLs, is sufficient to sequester enough cholesterol for dauer exit.

As mentioned above, cholesterol is the source of DAs, which in turn play an essential role in transcriptional reprogramming between quiescence and growth states. In its DA-bound state, the nuclear hormone receptor DAF-12 promotes reproductive development and suppresses dauer formation, therefore facilitating dauer exit[12]. We asked whether supplementation with DA can influence the impaired dauer exit phenotype in *scl-12&13;daf-2*. Indeed, 200 nM DA fully restored recovery rate and growth during dauer exit at 15 °C (Fig. 3a, b). This leads to the conclusion that proper localization and transport of cholesterol in dauer is of importance for biosynthesis of DAs during exit.

We next studied whether cholesterol sequestration in dauer larvae is sufficient for dauer exit. We grew *daf-2* animals at 25 °C on either control conditions (+chol) or strictly sterol-free (−chol) conditions, as previously described[13], and let them develop into dauer. We then induced dauer exit in the presence (+chol) or absence of cholesterol (−chol) and scored recovered worms after 48 h (described in Fig. 3c). As expected, worms recover normally when kept in +chol conditions at all times (entry +chol/exit +chol) and display impaired recovery under −chol conditions (entry −chol/exit −chol). Notably, in −chol/−chol conditions, worms show a recovery rate of about 40%, the same as in *scl-12&13;daf-2* mutants (Fig. 3d). Remarkably, however, we found that *daf-2* grown on +chol exit normally without cholesterol (entry +chol/exit −chol), indicating that the presence of accumulated internal cholesterol during pre-dauer development is sufficient for dauer exit. This was further confirmed by the fact that the exit of animals that were grown under −chol conditions is impaired even upon the addition of external cholesterol (entry −chol/exit +chol). Under these conditions, worms grow at the same rate as worms that were kept under sterol-free conditions at all times (entry −chol/exit −chol). In *scl-12&13;daf-2* mutants,

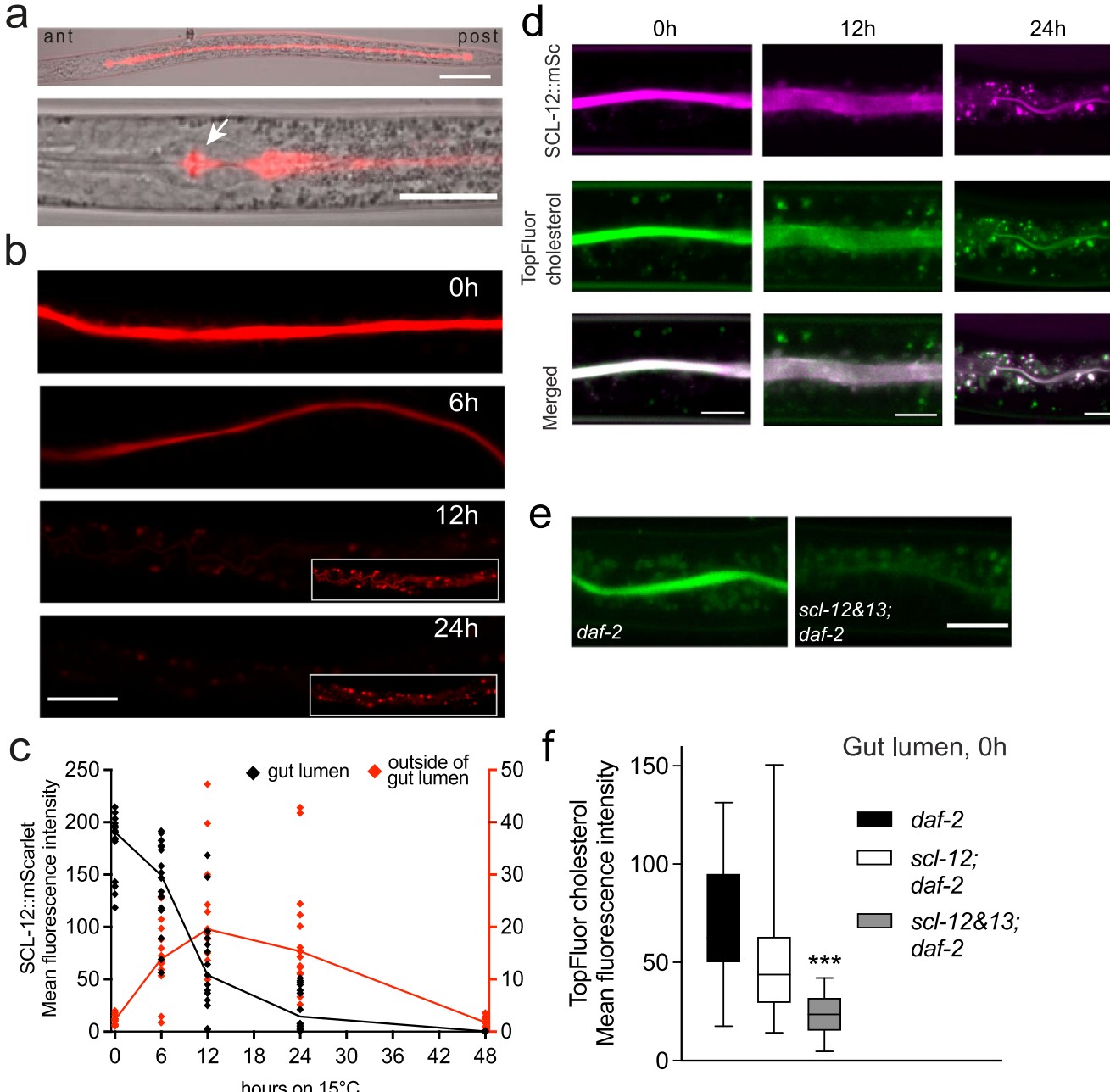

**Fig. 2 SCL-12 accumulates in the gut during the dauer state where it sequesters cholesterol. a** SCL-12::mScarlet reporter dauer larva, whole animal (top), head region (bottom), posterior (post), and anterior (ant) directions are marked. Scale bar top: 50 μm, bottom: 25 μm. **b** SCL-12::mScarlet;*daf-2* reporter dauer (0 h) and while exiting dauer at 6, 12, and 24 h after temperature switch to 15 °C. 12 h and 24 h insets are brightness adjusted images. Scale bar: 20 μm. **c** Mean fluorescence intensity of SCL-12::mScarlet;*daf-2* reporter dauers (0 h) and while exiting dauer after temperature switch to 15 °C. Black: Fluorescence in the gut lumen. Red: Fluorescence outside of the lumen. Means and individual values of at least 14 individual worms (48 h: 7 worms). **d** SCL-12::mScarlet reporter animals (SCL-12::mSc; magenta) and fluorescent TopFluor cholesterol (green) in the gut of a dauer larva (0 h; left panels), 12 and 24 h after induction of dauer exit (middle and right panels). Merged channels show a co-localization of both fluorophores. Scale bars: 10 μm. **e** *daf-2* and *scl-12&13;daf-2* dauers fed with TopFluor cholesterol prior to entering the dauer state, representative images. Scale bar: 15 μm. **f** Mean fluorescence intensity of TopFluor cholesterol in the gut lumen from E, *daf-2* (black) and *scl-12&13;daf-2* (gray), and a single-mutant *scl-12;daf-2* (white). Mean of 28 individual worms, unpaired two-tailed *t*-test with Welch's correction showed p < 0.0001 between *daf-2* and *scl-12&13;daf-2* and p = 0.08 between *daf-2* and *scl-12;daf-2*.

however, this phenomenon could not be observed, as all animals showed the same impaired recovery independently of cholesterol conditions, mimicking the effect of lacking dietary cholesterol in pre-dauer animals (entry −chol/exit −chol; Fig. 3d). Therefore, a sequestered, internal pool of cholesterol is a prerequisite for proper dauer exit.

**Cholesterol is transported from the gut lumen to lysosomes via the endocytic pathway and then further transported to the epidermal region.** As shown above, SCL-12 and cholesterol move from the gut lumen to punctate structures during dauer exit, where SCL-12 is degraded. We hypothesized that these structures could be lysosomes. To test this, we co-labeled *daf-2* dauer larvae

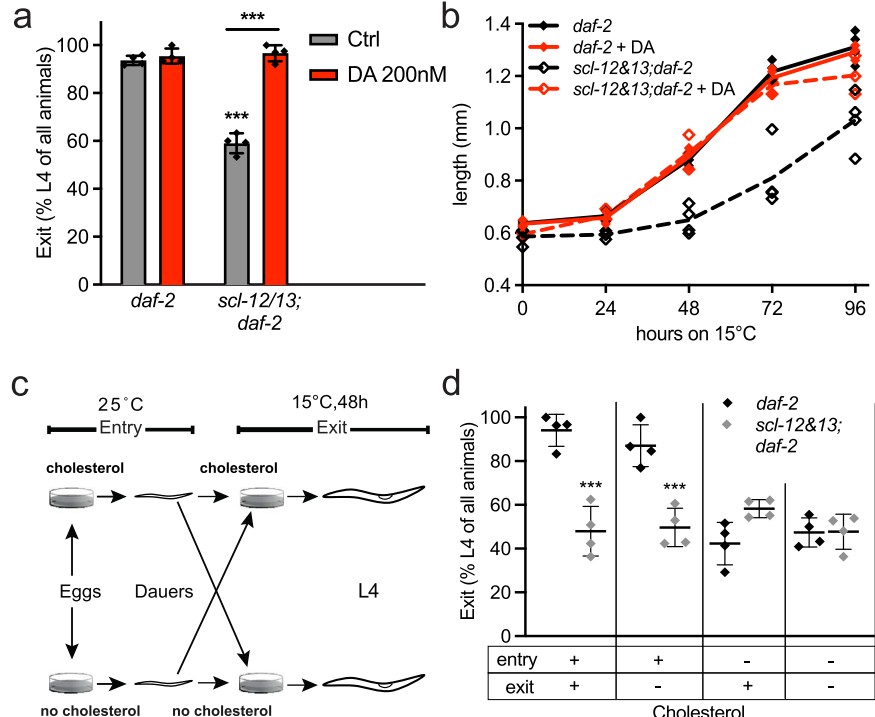

**Fig. 3 *scl-12&13* mutations lead to cholesterol-dependent impaired dauer exit. a** Percentage of *daf-2* and *scl-12&13;daf-2* populations that develop into L4 larvae 48 h after dauer recovery was induced by temperature switch to 15 °C, control (ethanol; black) and when treated with 200 nm (25S)-Δ7-dafachronic acid (DA; red). Average ± SD of four independent experiments with a minimum of 15 worms, two-way ANOVA with post hoc Sidak's multiple comparisons test comparing *daf-2* Ctrl versus *scl-12&13;daf-2* Ctrl showed *p* < 0.0001 and *scl-12&13;daf-2* Ctrl versus *scl-12&13;daf-2* DA *p* < 0.0001. **b** Growth/length in mm of *daf-2* and *scl-12&13;daf-2* after induction of dauer exit by temperature switch and the effect of DA thereon. Individual values of four independent experiments with a minimum of 10 worms, two-way ANOVA comparing *daf-2* Ctrl versus *scl-12&13;daf-2* Ctrl showed *p* = 0.0001, and *scl-12&13;daf-2* Ctrl versus *scl-12&13;daf-2* DA *p* < 0.0001. **c** Scheme for the experimental procedure in (**d**). **d** Percentage of *daf-2* (black) and *scl-12&13;daf-2* (gray) populations that develop into L4 larvae 48 h after dauer recovery was induced by temperature switch to 15 °C. Worms were grown with or without cholesterol (+ or -) from egg to dauer ("Entry") and with or without cholesterol from dauer to L4 while exiting dauer ("Exit"). Average ± SD and individual values of four independent experiments with a minimum of 15 worms, two-way ANOVA with post hoc Sidak's multiple comparisons tests comparing *daf-2* chol+/chol+ versus *scl-12&13;daf-2* chol+/chol+ showed *p* < 0.0001, *daf-2* chol+/chol− versus *scl-12&13;daf-2* chol+/chol− *p* = 0.0002, *daf-2* chol-/chol + versus *scl-12&13;daf-2* chol-/chol+ *p* = 0.0432 as well as *daf-2* chol+/chol+ versus *daf-2* chol−/chol+ *p* = 0.0004 and *daf-2* chol+/chol+ versus *daf-2* chol−/chol− *p* = 0.0001.

with TopFluor cholesterol (green) and Lysotracker to label the lysosomes (red). 24 h after exit, there is a strong overlap between these markers (Fig. 4a). After 48 h, Lysotracker and TopFluor cholesterol no longer co-localize and TopFluor fluorescence is seen in concentrated blobs in the epithelial region, which could be lipid droplets. This transport across the pseudocoelomic cavity to the epithelial region could mean that cholesterol undergoes further spatial redistribution and eventually serves for DAF-9-dependent DA synthesis in the hypodermis[29]. To confirm this result, we crossed the SCL-12::mScarlet;*daf-2* reporter strain (red) into the bona fide lysosomal marker LMP-1::GFP (green) and found a partial overlap of the fluorescent signals during dauer exit (Fig. 4b), consistent with SCL-12 and cholesterol moving to the lysosomes during dauer exit.

**Intact lysosomes are essential for dauer *exit*.** Next, we asked which pathway could be responsible for the transport of SCL-12 and SCL-13 and cholesterol to the lysosomes. For this, we analyzed dauer exit in *daf-2* worms fed with dsRNAs corresponding to key proteins involved in pathways such as apical trafficking, ESCRT, and endocytic/lysosome-related organelle pathways, some of which are shown in Fig. 5a[30–32]. Some of the tested RNAis caused lethality before the animals reached the dauer state, such as *rab-5* and *vps-32.2*, and therefore had to be excluded from

our analysis. From all tested RNAis, only *rab-7* showed a strong phenotype of delayed growth and generally impaired dauer exit at all measured time points (Figs. 5a and S3a). RAB-7 is a late endosomal GTPase responsible for lysosomal fusion[33]. Accordingly, we found that *rab-7* RNAi treatment destroys the lysosomal structures in dauers, which we visualized using a strain containing the lysosomal reporter LMP-1::GFP;*daf-2*. LMP-1 in the mock treatment appeared in the lysosomal membrane. During *rab-7* RNAi treatment, however, LMP-1 seems to be increased in quantity, but diffusely distributed throughout the cytosol or aggregated in larger structures (Fig. 5b). Investigating TopFluor cholesterol distribution in *rab-7* RNAi treated dauers, we found that there is less cholesterol stored in the gut, but significantly more in the peripheral tissues, probably in lipid droplets, compared to EV (Fig. S3b, S3c). Given the strong impairment of dauer exit caused by *rab-7* RNAi, we propose that the formation of functional lysosomes is an essential process for dauer exit. To further test this possibility, we treated *daf-2* animals with RNAi against the gene coding for HLH-30, the *C. elegans* ortholog of TFEB, the key lysosomal transcription factor regulating lysosomal biogenesis[34], and the lysosomal membrane protein LMP-1, an ortholog of mammalian LAMP1, responsible for lysosomal integrity. To assess the condition of lysosomes, we stained dauers that were treated with the respective RNAis with Lysotracker dye. We found a robust increase in fluorescence, as well as fewer and

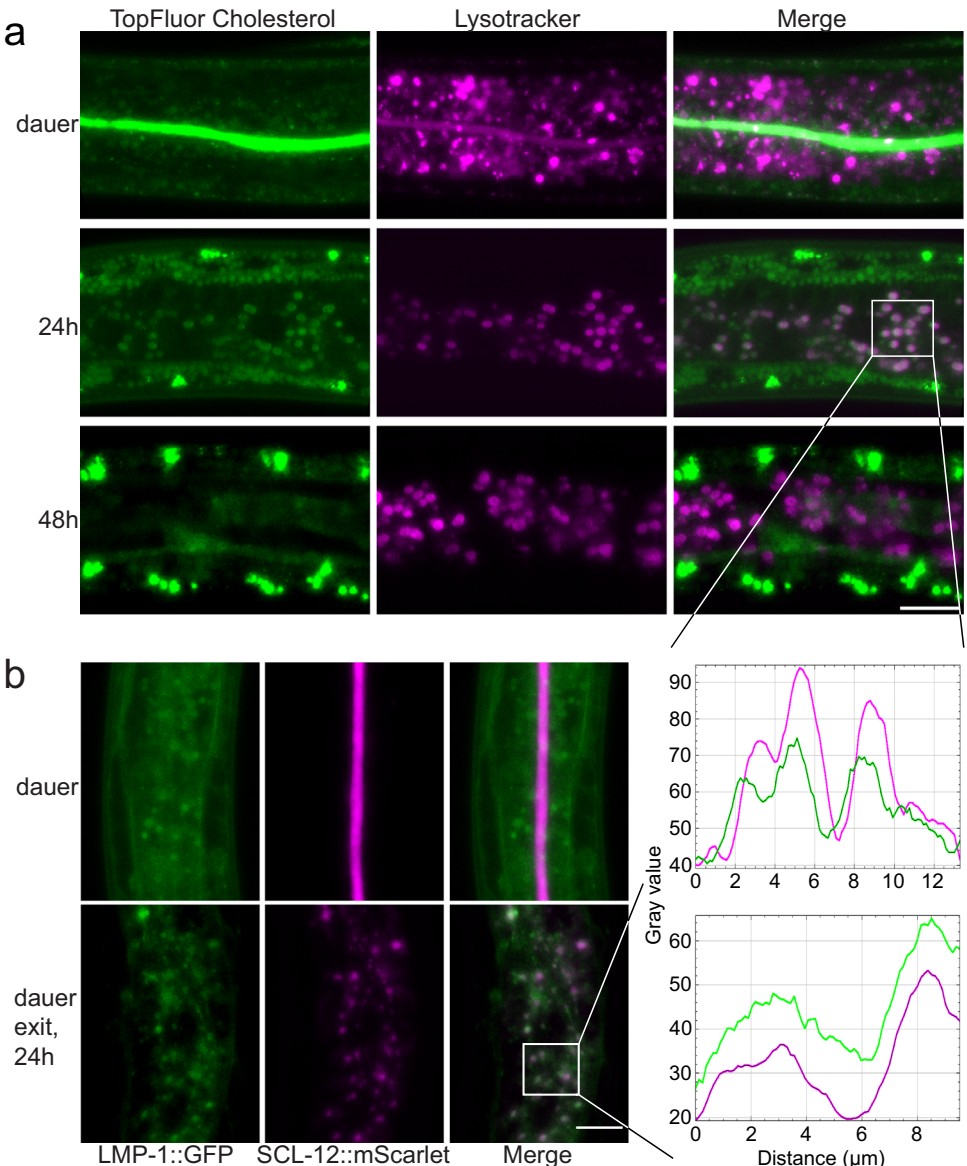

**Fig. 4 Cholesterol is transported from the gut lumen to intestinal lysosomes with SCL-12 and then further to the epidermal region. a** *daf-2* dauers co-labeled with TopFluor cholesterol (green) and Lysotracker dye (magenta; top panels), and 24 and 48 h after induction of dauer exit via temperature switch. Scale bar: 15 µm. The profile (fluorescence intensity in space) of both fluorophores in intestinal cells is similar at 24 h of dauer exit, indicating co-localization. **b** LMP-1::GFP;SCL-12::mScarlet;*daf-2* dauers (top panels) and 24 h after induction of dauer exit (bottom panels). Scale bar: 10 µm. Lysosomal LMP-1::GFP (green) and SCL-12::mScarlet (magenta) show a similar profile at 24 h of dauer exit, indicating co-localization.

bigger lysosomes in the absence of *hlh-30*, which has been previously reported[35]. *lmp-1* RNAi resulted in reduced lysotracker fluorescence with barely visible lysosomes, which could be due to loss of lysosomal integrity (Fig. S3d). Dauer exit is also slightly delayed in worms treated with *lmp-1* and *hlh-30* RNAis (Fig. S3e).

We assumed that the lack of functional lysosomes impairs sterol transport and thus its availability to DAF-9, an enzyme producing DAs. As shown above (Fig. 3a, b), DA fully restored the rate of growth during the dauer exit of *scl-12&13;daf-2*. To our surprise, DA had no effect on exit and growth of *rab-7* RNAi-treated animals (Fig. 5c). However, when these animals were fed a diet containing an about 75 times higher cholesterol concentration than in standard culture conditions (1 mM versus 13 µM), their growth deficit during dauer exit could be partially rescued (Fig. 5d). This suggests that the transport of cholesterol out of the gut lumen and to the lysosomes plays a role beyond the proper delivery to DAF-9 to form DAs.

**Cholesterol activates mTORC1 at the lysosomes to boost growth during dauer exit**. We next hypothesized that cholesterol delivered to the lysosomes by SCL-12 and SCL-13 during dauer exit might activate mTORC1 to boost protein synthesis and growth, as shown before in mammalian cells[36]. To test this hypothesis, we studied a loss-of-function mutant condition for mTORC1. Since mutations in or RNAi against RAPTOR/DAF-15 in *C. elegans* arrest development at the L3 stage, we exploited the auxin-inducible degron (AID) system[37] to generate a conditional knock-out of DAF-15 (*daf-15::mNeonGreen::AID*). We did not consider the TOR kinase LET-363 further because it is also part of mTORC2. AID relies on the exogenous expression of the plant F-Box protein TIR1, which mediates the depletion of the degron-tagged targets upon exposure to auxin. We expressed *daf-15::mNeonGreen::AID* in *ieSi57*(P*eft-3*::TIR1::mRuby) animals to knock-out DAF-15 in the soma (hereafter referred to as *daf-15::mNG::AID;TIR1*). We then crossed the resulting worms into

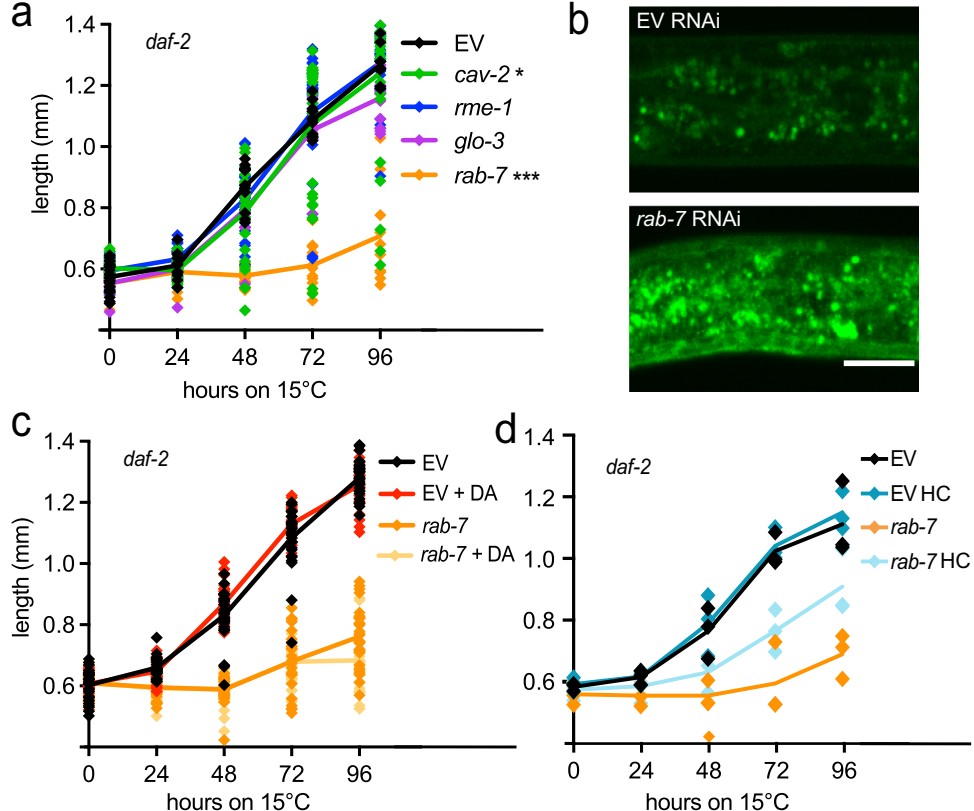

**Fig. 5 Intact lysosomes are essential for dauer exit. a** Growth/length in mm in *daf-2* exiting dauers induced by temperature switch to 15 °C on RNAis against factors of the apical trafficking and endocytic/lysosomal pathways, *rab-7* (orange), *cav-2* (green), *rme-1* (blue), and *glo-3* (purple) compared with mock treatment (EV: empty vector L4440; black). Average and individual values of 1 experiment with a minimum of 15 worms per condition, two-way ANOVA with post hoc Sidak's multiple comparisons test comparing EV versus *rab-7* showed $p < 0.0001$ and significant difference between the groups at all time points, EV versus *cav-2* only 48 h was significantly different between the groups ($p < 0.0001$), and EV versus *glo-3* only 96 h was significantly different between the groups ($p = 0.0016$). **b** Representing images of dauers of the lysosomal reporter LMP-1::GFP;*daf-2* treated with EV and *rab-7* RNAi. **c** Growth/length in mm in *daf-2* exiting dauers on EV and *rab-7* RNAi, treated with 200 nM DA (red/light orange), compared to solvent control (black/ orange). Average and individual values of 1 experiment with a minimum of 25 worms per condition, two-way ANOVA with post hoc Tukey's test comparing EV vs. *rab-7* showed $p < 0.0001$, EV vs. *rab-7* + DA $p < 0.0001$. *rab-7* vs. *rab-7* + DA was not significant. **d** Growth/length in mm in *daf-2* exiting dauers on RNAi against *rab-7* compared to EV, treated with 1 mM cholesterol (high cholesterol/HC; turquoise), compared to standard growing conditions that comprise 13 µM cholesterol (black/orange). Individual values of three independent experiments with a minimum of 30 individual worms per condition, two-way ANOVA with post hoc Tukey's test comparing EV vs. *rab-7* showed $p = 0.0056$, EV vs. *rab-7* HC $p = 0.0363$, and *rab-7* vs. *rab-7* HC $p = 0.002$.

the *daf-2* background to easily force dauer formation at 25 °C. As reported before in non-dauer larval stages, we found that DAF-15 is localized to the lysosomes in dauer larvae (Fig. S3a)[17]. To test the efficiency of AID, we exposed *daf-15::mNG::AID;TIR1* to 400 µM auxin, and found that the fluorescent tag of DAF-15 mostly disappears within one hour (Fig. S3b). Furthermore, animals synchronized as L1 larvae on auxin-containing plates arrest at the L3 stage, as reported in *daf-15* deletion mutants (Fig. S3c)[38].

We next examined exit in both *daf-15::mNG::AID;TIR1;daf-2* dauers and *daf-15::mNG::AID;TIR1* starvation dauers. Treatment with auxin leads to a strong decrease in growth compared to the control (Figs. 6a and S4d—shown in both *daf-2* and wt background), and animals never transit into L4 larvae. However, oxygen consumption during exit in auxin-treated worms increased concomitantly with exit, similar to control animals (Fig. S4e). Thus, DAF-15-dependent growth and mitochondrial activation/metabolic switch toward OxPhos are not coupled. This could in principle explain the premature death of the pool of animals that exit dauer on auxin plates: Fat deposits are metabolized and the resulting energy is wasted, without being usefully channeled to support growth (Fig. 6b). We tested this hypothesis by staining the fat storage in exiting dauers with

NileRed, a lipophilic dye. At 4 days after induction of exit, there is a tendency towards smaller fat deposits in *daf-15::mNG::AID;-TIR1;daf-2* treated with auxin, which becomes significant after 10 days (Fig. S4f, S4g).

Next, we aimed to investigate whether the cholesterol and DAF-15/Raptor work together to promote growth in vivo. Indeed, several lines of evidence supported this idea. First, we exploited the fact that *daf-15::mNG::AID;TIR1* arrest at the L3 state upon auxin incubation in L1 larvae. Worms were arrested on either cholesterol-containing or depleted plates (scheme in Fig. 6c). After the removal of auxin, the animals were allowed to recover in the presence or absence of cholesterol. *daf-15::mNG::AID;TIR1* could exit the auxin-induced arrest and restart growth under conditions where cholesterol was present during the whole experiment (+/+chol), where cholesterol was present only during the auxin treatment (+/−chol) or after removal of auxin (−/+chol). In contrast, worms grown without cholesterol (−/−chol) never resumed growth (Fig. 6d). These data indicate that DAF-15/mTORC1 can be activated only in the presence of cholesterol.

Above, it was shown that the growth deficit of animals treated with *rab-7* RNAi during dauer exit could be rescued with high

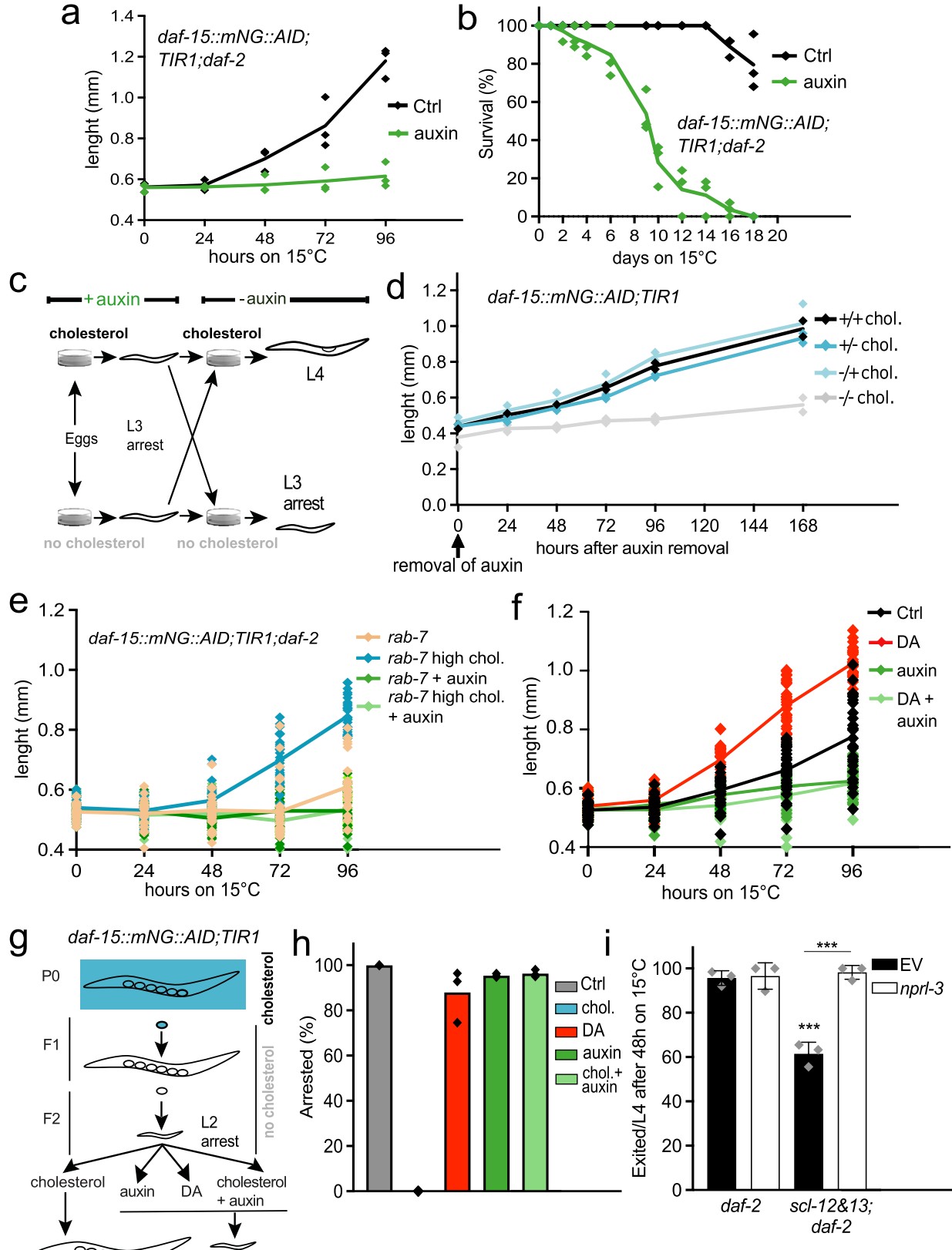

doses of cholesterol, but not DA (Fig. 5c, d). When DAF-15 is degraded, however, the effect of high cholesterol doses on *rab-7* RNAi treated dauers is abolished (Figs. 6e, S4h). Consistently, growth of *daf-15::mNG::AID;TIR1;daf-2* dauers that harbor an additional deletion of *scl-12&13* cannot be rescued by DA when DAF-15 is degraded on auxin plates (Fig. 6f; compare with

Fig. 3a, b). Thus, the activity of DAF-15 requires intact lysosomes and the presence of cholesterol but is not connected to the DA pathway.

Our data so far indicated that SCL-12/13-mediated cholesterol transport from the gut lumen to the intestinal lysosomes promotes growth partly through activation of mTORC1.

**Fig. 6 Cholesterol activates mTORC1 at the lysosomes to boost growth during dauer exit. a** Growth/length in mm in *daf-15::mNeonGreen::AID;TIR1;daf-2* exiting dauers induced by temperature switch treated with 400 µM auxin (green) or solvent control (ethanol; black). Average and individual values of 3 independent experiments with a minimum of 30 worms per condition, two-way RM ANOVA with Geisser-Greenhouse correction showed $p = 0.0018$. **b** Survival of *daf-15::mNeonGreen::AID;TIR1;daf-2* exiting dauers treated with auxin or solvent control. Average and individual values of one experiment with three biological replicates. Log-Rang analysis of the solvent control group compared to the auxin-treated group showed $p < 0.0001$. **c** Scheme of experimental procedure in (**d**). **d** *daf-15::mNeonGreen::AID;TIR1* grown on 400 µM auxin and with cholesterol (+chol.) or without cholesterol (−chol.) until arrested as L3. Auxin was removed (0 h) and worms started re-growth on +chol. or −chol. Average and individual values of 2 independent experiments with a minimum of 30 worms per condition, two-way ANOVA with post hoc Tukey's test comparing +/+chol with −/−chol showed $p = 0.0007$. **e** Growth/length in mm in *daf-15::mNeonGreen::AID;TIR1;daf-2* exiting dauers on RNAi against *rab-7*, treated with 1 mM cholesterol (turquoise) or 13 µM cholesterol (black/orange) and with (green) or without 400 µM auxin (light green). Average and individual values of one experiment with a minimum of 29 worms per condition, two-way ANOVA with post hoc Tukey's test comparing *rab-7* with *rab-7* high chol. showed $p < 0.0001$, *rab-7* high chol. with *rab-7* high chol. + auxin $p < 0.0001$. EV controls see Fig. S4h. **f** Growth/length in mm in *daf-15::mNeonGreen::AID;TIR1;scl-12&13;daf-2* exiting dauers on 200 nM DA (red), auxin (green) and a combination of DA and auxin (light green), compared to solvent control. Average and individual values of one experiment with a minimum of 24 worms per condition, two-way ANOVA with post hoc Tukey's test comparing Ctrl vs. DA showed $p < 0.0001$, Ctrl vs. auxin $p < 0.0001$, Ctrl vs. DA + auxin $p < 0.0001$, and DA vs. DA + auxin $p < 0.0001$. **g** Scheme of experimental procedure in (**h**). **h** *daf-15::mNeonGreen::AID;TIR1* were grown without cholesterol until they were arrested in the F2 generation. Graph shows the percentage of the population that stays in the arrested state upon the addition of cholesterol (0%), 200 nM DA (red), 400 µM auxin (green), and cholesterol together with auxin (light green), compared to the solvent control. Average and individual values of three independent experiments with a minimum of 40 individual worms per condition. Only cholesterol is fully able to rescue the arrest (3 times an experimental value of 100% for Ctrl and 3 times 0% for cholesterol). An unpaired *t*-test comparing Ctrl and DA treatment showed $p = 0.1497$, and comparing cholesterol and DA $p = 0.0002$. **i** Exited *daf-2* and *scl-12&13;daf-2* worms/L4 after 48 h of induction of dauer exit treated with EV and *nprl-3* RNAi (white). Average and individual values of 3 independent experiments with a minimum of 25 worms per condition each, unpaired two-tailed *t*-test between *daf-2* EV and *scl-12&13;daf-2* EV showed $p = 0.0007$ and between *scl-12&13;daf-2* EV and *scl-12&13;daf-2 nprl*-3 $p = 0.0005$.

To address this hypothesis further, we used an RNAi against the gene coding for Nitrogen Permease Regulator Like 3 (*nprl-3*), which has been shown to activate mTORC1 constitutively[39]. The NPRL-2/NPRL-3 complex, similar to its mammalian orthologue GATOR1, represses mTORC1, and loss of either NPRLs causes a robust mTORC1 hyperactivation[40]. If *scl-12&13* mutants partially arrest during dauer exit and this is due to a lack of mTORC1 activation, the downstream hyperactivation of mTORC1 should restore their ability to exit the dauer state. Indeed, 100% of the *scl-12&13;daf-2* animals grown on *nprl-3* RNAi bypassed the partial arrest during dauer exit (Fig. 6i). This indicates that SCL-12 and SCL-13 activate mTORC1 during dauer exit by delivering cholesterol to the lysosomes, which is necessary to boost growth after remaining in a quiescence state through the activation of mTORC1 signaling.

Previously, we demonstrated that worms grown under sterol-depleted conditions will arrest as L2* larvae in the F2 generation, and continue development when cholesterol (but not DA) is restituted to the food[13,14]. Strikingly, we found that the activity of mTORC1/DAF-15 is essential to restart development also in this case. *Daf-15::mNG::AID;TIR1* worms grown on auxin plates could not exit the arrested L2* stage after the switch to cholesterol (Fig. 6g, h).

To conclude, under all conditions tested, exit from dauer and growth in *C. elegans* is only possible if DAF-15 and cholesterol are both present. Hence, it is very likely that they act within the same cellular process.

## Discussion

In this report, we present data showing that the mobilization of cholesterol from internal stores is the key event during the transition from dauer state to growth in *C. elegans*. During dauer formation, cholesterol is sequestered in the gut lumen by sterol-binding proteins of the SCL family. Upon exit signal, the complex of SCL-proteins and cholesterol is endocytosed into the intestinal lysosomes. Here, the proteins of the complex are degraded and cholesterol is released. Now, it can participate in two essential processes: (1) Transcriptional reprogramming via binding to the nuclear hormone receptor DAF-12 and (2) Activation of mTOR

and thus transition to full-scale protein synthesis and consequently growth.

Our data show that *C. elegans* do not only rely on the environment to provide exogenous cholesterol but store it to have immediate access when exiting dauer. This is an elaborated way of sequestration and compartmentalization in the gut lumen that is most likely a mechanism to minimize the signaling properties of cholesterol during a quiescence state. As previously shown, the transcriptional switch utilizing cholesterol that is needed to exit dauer involves the cytochrome P450 DAF-9 and nuclear hormone receptor DAF-12[11,41]. Cholesterol serves as a precursor for DA production to DAF-9, which leads to DA-binding of DAF-12 and a transcriptional switch toward reproductive development. The storage of cholesterol during the dauer stage in the gut lumen as opposed to lipid droplets could serve as a means to isolate it from the site of DA synthesis, the hypodermis. Leakage of precursors could result in undesired DA synthesis when environmental cues still suggest the dauer mode. Interestingly, in 2010, we found that the lumen of the *C. elegans* dauer gut is entirely filled with a compact multilamellar material, visible in electron microscopy images as concentric circles in cross-sectional cuts[42]. The data in the present study prompts the possibility that this material contains complexes of SCL proteins and cholesterol.

Cholesterol sequestration and trafficking by SCL-12 and SCL-13 described here in *C. elegans* might have important parallels in humans, in particular in male fertility. The human orthologues of SCL-12 and SCL-13 are glioblastoma pathogenesis-related 1 (GLIPR2/GAPR1) and cysteine-rich secretory proteins (CRISP2; CRISP3). The latter are found in the male reproductive tract, especially in sperm. Notably, cholesterol, which is most likely bound to these proteins, needs to be released from the spermatozoa membrane, inducing a variety of processes all needed for successful capacitation (i.e. the physiological changes needed to penetrate and fertilize an oocyte)[24,43]. Interestingly, we previously have shown that the reproductive system of *C. elegans*, in particular the sperm, is also enriched in cholesterol[44]. Here, it might have an additional structural function for the initial rapid germline development upon dauer exit. Along these lines, the sequestration and absence of cholesterol during dauer might be an effective way to maintain germline quiescence, which is

necessary for reproductive fitness in post-dauer animals[45]. Notably, SCL-12 is not only found in dauers but also in male *C. elegans*[46].

Our data show a cholesterol-driven activation of mTOR at the lysosomes as a modulator of metabolic and developmental plasticity. It is widely accepted that lysosomes do not only serve as recycling centers for amino acids, but also as a hub for intracellular distribution of cholesterol and other nutrients. In fact, the lysosomes serve as a sorting station for dietary cholesterol in eukaryotes. In mammalian cells, low-density lipoproteins (LDLs) carrying cholesterol enter the lysosomes via endocytosis and are disassembled in the lumen. From there, a sterol transport system containing the Nieman-Pick C1 (NPC1) and NPC2 proteins that are localized at the lysosomes binds free cholesterol and mediates its transport to a variety of cellular compartments, such as the plasma membrane and the endoplasmic reticulum[47]. A variety of signaling processes are initiated at the lysosomal surface, including growth mediated by mTOR. Hormones, growth factors, and nutrients including glucose, amino acids, and a variety of lipids such as cholesterol can activate the mTOR pathway, which occurs in general when environmental conditions are favorable. Subsequently, mTOR regulates transcription factors such as SREBP to drive lipid biosynthesis and induces ribosome biogenesis and protein synthesis, ultimately leading to cellular growth. When conditions are averse, mTOR signaling is suppressed, resulting in the inhibition of global protein synthesis and substantial energy savings[48]. In light of this paradigm, it makes sense that mTOR signaling is kept at lower levels during the dauer stage and needs an activation signal at the time of exit.

It is also important to mention that sterols, for instance sex hormones, are crucial signaling molecules in a variety of cancers. Cholesterol and its derivatives are not only precursors of many sterols with potentially signaling properties, but also activate mTOR directly in cancer cells[49]. Recently, it has been shown that increased cellular cholesterol can amplify the progress from non-alcoholic fatty liver disease to hepatocellular carcinoma[50], and blocking cholesterol synthesis and uptake might have inhibitory effects on tumor formation and growth[51]. The data presented here therefore might be of importance far beyond metabolic and developmental plasticity, such as in the emerging field of finding drug targets in cholesterol metabolism and transport for cancer treatment.

mTORC1 in *C. elegans* is localized at the lysosomal membrane at all times during the larvae stages including dauer, as shown in this study and by others[17]. In mammals, the translocation of cytosolic mTORC1 to the lysosome membrane is an important step of its activation[52], and mTORC1 activation by cholesterol requires the lysosomal GPCR-like protein LYCHOS as well as the transmembrane protein SLC38A9[53–55]. Additionally, NPC1 binds to SLC38A9 and is capable of inhibiting mTORC1 signaling[36]. Whereas a LYCHOS gene cannot be found in nematodes, both SLC38A9 and NPC1 have orthologues in *C. elegans*, F13H10.3 and NCR-1/NCR-2, respectively[56,57]. Whilst *ncr-2;ncr-1* knock-out mutants arrest as dauers instead of forming fertile adults as a result of decreased DA production[12,57], neither NCR-1/NCR-2 nor F13H10.3 in *C. elegans* seems to be involved in mTORC1 activation as the RNAi-induced knock-down has no effect on dauer exit in *daf-2* (Fig. S5). It is therefore likely that an alternative molecular mechanism mediates cholesterol/mTORC1 signaling through the lysosomal membrane that has yet to be discovered.

Two decades ago we postulated that *C. elegans* requires too little cholesterol for it to have a major role in membrane structure and thus should have only signaling functions[58]. Work from several laboratories corroborated that cholesterol is a precursor of DA, a steroid hormone, and thus regulates the transition between dauer state and reproduction[12,13]. Another aspect of its function, involvement in growth, however, remained enigmatic. Namely, if being kept under completely sterol-free conditions, the first generation of worms can still grow from eggs to adulthood. In the second generation though, they develop into dauer-like larvae with incomplete molting (so-called L2*)[13]. This larval stage can be rescued by the addition of cholesterol but not DA[14], although our results show that DA might, to a very weak extent, be able to rescue the L2* arrest. Interestingly, also treatment with phosphorylated glycosphingolipids leads to the exit of the L2* arrest via removing a lysosomal blockage, as they rescue *ncr-1;ncr-2* mutants[59]. It has to be further investigated, however, if this happens downstream of SCL-12 and SCL-13, or if this is an SCL-12- and SCL-13-independent mobilization of cholesterol.

Our data presented here answers the question stated in "Why do worms need cholesterol?"[58]: For the production of DA and for the activity of mTOR, although some additional aspects of cholesterol action may be unraveled in the future.

## Methods

**Chemicals**. All chemicals came from Sigma-Aldrich (Taufkirchen, Germany) unless otherwise specified.

***C. elegans* maintenance and strains**. *C. elegans* were propagated as described here[60]. Briefly, worms were grown on NGM agar plates that were streaked with *Escherichia coli* NA22 as a food source at 15 °C. The N2 Bristol wildtype strain (wt), *daf-2(e1370)*, the lysosomal reporter LMP-1::GFP*(vkIs2882)*, the TIR1$_{soma}$ expressing strain P$_{eft-3}$::TIR1::mRuby *(ieSi57)* and the *E. coli* strain NA22 were provided by the Caenorhabditis Genetics Center (CGC) at the University of Minnesota.

The strains *scl-12&13* (double mutants) and *scl-12;daf-2* (single mutants), SCL-12::mScarlet and *daf-15::mNeonGreen::AID* were created for this study using the CrisprCas9 system. The compound mutant strains *scl-12&13;daf-2*, SCL-12::mScarlet;-*daf-2*, LMP-1::GFP;*daf-2*, LMP-1::GFP;SCL-12::mScarlet, LMP-1::GFP;SCL-12::mScarlet;*daf-2*, *daf-15::mNeonGreen::AID;TIR1* and *daf-15::mNeonGreen::AID;TIR1;daf-2* were generated by crossing. All genotypes were confirmed by PCR.

Worms were synchronized by hypochlorite treatment of gravid adults[61]. Wt dauers were obtained by SDS treatment of overcrowded, starved worm populations. These populations were washed from the plates and treated with 1% SDS for 30 min, shaking at 30 °C. After that, worms were washed 3× with ddH$_2$O and spread on NGM plates, where surviving dauers separated from dead non-dauers. For *daf-2(e1370)* dauers, eggs were placed on NGM plates and grown at 25 °C for 72 h, in contrast to 15 °C, the restrictive temperature that was used for maintenance of all mentioned strains.

**Growth in sterol-free and high-cholesterol conditions**. The growth on the sterol-free medium was performed as previously described[13]. Briefly, a sterol-depleted medium was obtained by substituting NGM agar with agarose that was extracted three times with chloroform to remove all traces of sterols. NA22 bacteria were grown in sterol-free DMEM medium, pelleted, and resuspended in M9 buffer. For the F2 arrest assay, worms were propagated on these plates for two consecutive generations. Subsequently, synchronized L1 larvae were grown at 25 °C for 72 h in the F2 generation until developmental arrest occurred.

High cholesterol conditions were prepared as described here[59]. Briefly, cholesterol powder was autoclaved and 1 ml *E. coli* suspension was added to 1 mg of cholesterol crystals. The bacterial/cholesterol suspension was then incubated for 1 h at 37 °C on a rotary shaker at 200 r.p.m. for the cholesterol to solve.

NGM-agar plates were then seeded with this bacterial/cholesterol suspension.

**DA treatment**. (25S)-Δ7–DA was produced in the laboratory of Prof. H.-J. Knölker at Technical University Dresden and dissolved in ethanol. The stock solution was added to the bacteria to a final concentration of 200 nM, calculated according to the volume of the NGM agar.

**2-Dimensional difference gel electrophoresis (2D-DIGE)**. Wt dauers were obtained by SDS treatment as described above. Dauers were then kept on NGM agar plates without bacteria lawn (Ctrl) or were placed on NGM plates with NA22 lawn, and exit via food introduction was induced for 4 h. After that, the animals were collected in ddH$_2$O, washed 3× and snap frozen in liquid nitrogen. Samples were lysed by five freeze–thawing cycles in an ultrasound water bath. Protein content was determined with an RC DC™ Protein Assay (BioRad, Germany). 50 µg of protein per sample was solved in urea lysis buffer and labeled with 250 pM CyDye DIGE Fluor dyes (GE Healthcare, Germany). Subsequently, 10 nM L-lysine was used to quench excess dyes, and samples were reduced in rehydration buffer (7 M urea, 2 M thiourea, 50 mM DTT, 4% CHAPS). Ampholytes (BioLytes pH 3–10, BioRad, Germany) were added to a total volume of 350 µl. The labeled protein mixture was transferred by passive rehydration for 24 h into an immobilized pH gradient strip (linear pH 3–10). After that, isoelectric focusing for 55–60 kVh in total was performed in a Protean IEF cell (BioRad, Germany) and the gradient strip was equilibrated in equilibration buffer (6 M urea, 50 mM Tris, 130 mM DTT, 2% SDS, 20% glycerol) for 10 min, before being loaded on a 20 cm wide 12% SDS–polyacrylamide gel. The proteins in the strip were separated by SDS–PAGE (200 V, 5 h) and the gel was imaged using a Typhoon 9500 fluorescence imager (GE Healthcare, Germany) at a resolution of 100 µm/pixel for Cy2 at 488 nm excitation, BP 520/40 emission filter, for Cy3 at 532 nm excitation, BP 580/30 emission filter, and for Cy5 at 633 nm excitation, BP 670/30 emission filter. After imaging, Coomassie blue staining was performed and spots of interest were cut out. The proteins in these gel spots were extracted and identified via geLC–MS/MS[62].

**Label-free LC–MS/MS quantification of *C. elegans* proteins**. All the reagents used in the experiments are of analytical grade. LC–MS grade solvents were purchased from Fisher Scientific (Waltham, USA); formic acid (FA) from Merck (Darmstadt, Germany), Complete Ultra Protease Inhibitors from Roche (Mannheim, Germany); Trypsin Gold, mass spectrometry grade, from Promega (Madison, USA). Other common chemicals and buffers were from Sigma-Aldrich. The samples for 0 h (dauer) and 24 h (exiting dauers) time points were produced in three biological replicates. Worms were collected from NGM plates with M9 buffer and washed twice, counted, transferred to lysis buffer, and snap-frozen in liquid nitrogen. The frozen worms were thawed on ice and crushed using a micro hand mixer (Carl Roth, Germany). The crude extract was centrifuged for 15 min at 13,000 rpm at 4 °C to remove any tissue debris and the clear supernatant was transferred to a fresh Protein Low-Bind tube (Eppendorf, Hamburg, Germany). The total protein content of the samples was estimated using Pierce BCA protein assay kit from Thermo Scientific (Rockford, USA), and 30 µg of total protein content was loaded to a precast 4–20% gradient 1 mm-thick polyacrylamide mini-gels from Anamed Elektrophorese (Rodau, Germany). Proteins were in-gel digested and analyzed by LC–MS/MS on a hybrid Q Exactive tandem mass spectrometer as described in ref. [63]. Peptide matching was carried out using

Mascot v.2.2.04 software (Matrix Science, London, UK) against *C. elegans* proteome downloaded from Uniprot (November 2020), to which common protein contaminants (e.g. human keratins and other skin proteins, bovine trypsin, etc.) have been added. The precursor mass tolerance and the fragment mass tolerance were set to 5 ppm and 0.03 Da, respectively. Fixed modification: carbamidomethyl I; variable modifications: acetyl (protein N terminus), oxidation (M); cleavage specificity: trypsin, with up to 2 missed cleavages allowed. Peptides having ions score above 15 were accepted (significance threshold $p < 0.05$). The label-free quantification and subsequent statistical analysis were performed using MaxQuant (v 1.6.0.16) and Perseus (v1.6.2.1), respectively. The raw data have been deposited to the ProteomeXchange Consortium via the PRIDE[64] partner repository with the dataset identifier PXD048087.

**In vitro sterol binding assay**. SCL-12 containing a C-terminal polyhistidine tag was expressed in Sf9 insect cells using the *Baculovirus* expression system and purified using standard nickel-based affinity chromatography followed by size-exclusion chromatography. In vitro binding was performed in a 96-well microassay plate with black walls and a clear bottom (Greiner Bio-One, Kremsmünster, Austria) in triplicates. The purified protein was solved in PBS at a concentration of 0.2 mg/ml, and 10× fatty acid binding buffer was added according to the individual volume to reach a final concentration of 20 mM Tris (pH 7.5), 30 mM NaCl, and 0.05% Triton X-100. 10 µl of this protein solution was incubated with different concentrations of TopFluor cholesterol (0, 10, 50, 100 nM in EtOH) for 1 h at room temperature, or pre-incubated with 50 or 100 nm cholesterol for 30 min. Meanwhile, 12 µl Q-Sepharose beads (Cytiva, Marlborough, USA) were prepared to separate the protein from the unbound ligand. Beads were washed 2× with lipid binding buffer, and then the sample was added for 1 h at room temperature. Beads with bound protein were washed 3× with clean binding buffer to remove unbound ligands. After that, fluorescence was measured with a Tecan Spark fluorometer (Männedorf, Switzerland).

**Microscopy**. For fluorescence microscopy, worms were placed on glass slides (Superfrost Plus by Thermo Scientific, Waltham, USA) mounted with 1.5% agarose pads, and 5 mM levamisole in M9 buffer was used for immobilization. Samples were covered with coverslips (0.17 ± 0.005 mm; Menzel Gläser). Fluorescence microscopy was conducted with a Zeiss LSM 880 upright microscope and Zeiss ×10/0.45 Plan-Apochromat/Air, 20x/0.8 Plan-Apochromat Air and ×63/1.3 LCI Plan-Neofluar, W/Glyc, DIC objectives. Dauer exit and growth determination were performed with an Olympus SZX16 stereomicroscope and a QImaging camera.

**Oxygen consumption assay (OCR)**. To measure the oxygen consumption of *C. elegans*, a Seahorse XFe96 system (Seahorse Bioscience, North Billerica, USA) was used as previously described[65]. Worms were grown on NGM plates for 72 h at 25 °C to reach the dauer state. After that, the worms were removed from the plates, washed three times with M9 buffer, and transferred to fresh plates at 15 °C to induce dauer exit. After the respective time of exit, worms were collected and washed at least three times to remove all bacteria, compounds, and debris. About 100 worms/well were pipetted into a 96-well Seahorse XFe assay plate and OCR was measured at a temperature of 25 °C. Subsequently, the worms were removed from the wells and BCA protein determination (Thermo Scientific™ Pierce™ BCA Protein Assay Kit) was performed and used for normalization. At least six

wells (biological replicates) were used for each strain and condition.

**Dauer survival, dauer exit survival and exit rate, and post-dauer fertility**. Dauer larvae were prepared by growing the worms on a solid medium at 25 °C as described above. After collection, dauers were washed three times with M9 buffer and subsequently transferred into 15 ml centrifuge tubes (Corning, NY, USA) containing 10 ml sterile M9 buffer supplemented with streptomycin (50 μg/ml) and nystatin (10 μg/ml) at a density of 500 worms/ml. The temperature was kept at 25 °C and tubes were under constant agitation. Survival rate was determined in percentage by counting alive and dead dauers in 100 μl aliquots approximately every 5–10 days. Tubes were opened and the medium was allowed to oxygenate every 2–3 days in between measurements.

For the survival of exiting worms, dauers were washed 3 times after collection, transferred onto fresh plates with low density, and kept at 15 °C. Worms were separated from their progeny and scored for dead animals every 2–3 days.

Dauer exit rate and post-dauer fertility determination were performed as follows. Dauers were removed from plates, washed three times, and 1 dauer per well was allowed to exit the dauer state on fresh 12-well plates at 15 °C. For the dauer exit rate, plates were screened every day for worms reaching the L4 state, which was counted as "exited". Worms that remained in dauer or died before reaching L4 were considered "non-exited". Animals that escaped were censored. For post-dauer fertility determination, plates were checked for eggs and live progeny every day following exit induction.

Experiments were started with at least double the number of dauers than planned to use for statistical analyses as a large number was expected to escape.

**TopFluor cholesterol, Lysotracker Red, and Nile Red labeling of C. elegans**. TopFluor cholesterol was resolved in ethanol at a concentration of 6.5 mM, mixed with bacteria solution, and spread on top of cholesterol-free NGM plates at a concentration of 6.5 μM, calculated using the volume of the NGM agar. Top-Fluor cholesterol plates were used instead of standard NGM plates to grow *daf-2*, *scl-12&13;daf-2*, and SCL-12::mScarlet;*daf-2* at 25 °C until they reached the dauer state. For dauer exit, worms were transferred onto standard NGM plates at 15 °C without TopFluor cholesterol to monitor the distribution of the cholesterol that was sequestered during the dauer state.

LysoTracker™ Deep Red dye (Invitrogen™, Carlsbad, USA) was resolved in DMSO at a concentration of 1 mM and mixed with *E. coli* to a final concentration of 1 μM in the bacteria solution. Lysotracker-containing bacteria were spread on standard NGM plates and allowed to dry in the dark. Eggs were put onto those plates and worms were allowed to enter dauer. Plates were kept in the dark to avoid photobleaching. When dauer exit was induced, the animals were transferred onto fresh standard NGM plates without dye.

For fat staining with Nile Red, a 1 mg/ml Nile Red stock solution in DMSO was prepared. 0.3 μl of this solution was mixed with 50 μl *E. coli* that were concentrated 10×, which was then used to seed one well of a six-well NGM agar plate and quickly dried under a laminar flow. Worms were transferred to these plates for staining the same day and incubated overnight.

Mean fluorescence intensity of TopFluor cholesterol, Lyso-Tracker™ Deep Red, Nile Red, and SCL-12::mScarlet was measured using the Mean gray value analysis tool in FIJI[66]. Fluorescence profiles were prepared using the FIJI Plot profile tool.

**RNAi/AID experiments**. RNAi experiments were performed as described here[67]. RNAi clones (*E. coli* HT115) of *cav-2*(C56A3.7), *rme-1*(W06H8.1), *glo-3*(F59F5.2), *rab-7*(W03C9.3), *hlh-30*(W02C12.3), *lmp-1*(C03B1.12), F13H10.3, *ncr-1*(F02E8.6), *ncr-2*(F09G8.4), and *nprl-3*(F35H10.7) were taken from the ORFeome RNAi library (Open Biosystems) and experimentally compared to an empty vector clone (L4440). Sequencing to confirm the correct clone was performed before usage. RNAi bacteria were spread and allowed to dry on NGM plates containing 1 mM isopropyl-ß-D-thiogalactopyranoside and 50 μg/ml carbenicillin. After a minimum of 24 h, plates were used for experiments.

Auxin treatment was performed as described here[37]. A 400 mM stock solution of the naturally occurring auxin indole-3-acetic acid (Alfa Aesar/Thermo Scientific Chemicals, Haverhill, USA) in ethanol was prepared and NGM plates were poured containing 400 μM auxin.

**Statistics and reproducibility**. Statistical analyses were performed with GraphPad Prism 8.0.1 for Windows and Prism 10.1.1 for macOS (San Diego, USA). For dauer exit survival analyses, JMP® 17.0.0. (SAS Institute Inc., Cary, USA) was used. Statistical significance was assumed if $p < 0.05$. Which statistical test was applied to determine the *p*-value of all experiments can be found in the legends of the respective figure. All further details describing the individual statistical analysis for each experiment can be found in the raw data files (https://doi.org/10.17617/3.Y9PFVW).

**Reporting summary**. Further information on research design is available in the Nature Portfolio Reporting Summary linked to this article.

## Data availability

All raw data can be accessed on the EDMOND data depository database under https://doi.org/10.17617/3.Y9PFVW. Mass spectrometry proteomics data have been deposited to the ProteomeXchange Consortium via the PRIDE partner repository with the dataset identifier PXD048087.

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

## Acknowledgements

We would like to acknowledge the members of the Kurzchalia laboratory for continuous helpful discussions and support in experimental work. Huge thanks to the following people and their contributions: Mihail Sarov and Dana Olbert from the Genetic

Engineering Facility of MPI-CBG for the help generating the mutant strains. Karin Crell and Friederike Thonwart from the Grill laboratory for sharing their RNAi library and growing clones for us. Aliona Bogdanova from the Protein Biochemistry Facility for purifying proteins. The Light Microscopy Facility and Jan Peychl for continuous support during imaging. We furthermore thank Marcos Gonzales, Richard Roy, as well as other members of the *C. elegans*, MPI-CBG, and Dresden scientific communities for their insights, guiding suggestions, and encouragement.

## Author contributions

K.S., S.P., and T.V.K. designed the experiments. K.S. conducted and analyzed all experiments but the following: D.K. and S.P. performed the 2D-DIGE experiment. B.K.R. and A.S. performed and analyzed the mass spectrometry analysis, for which K.S. provided the samples. All authors discussed the results. S.P. and T.V.K. supervised the study. J.R. and T.V.K. financed the work. K.S. and T.V.K. wrote the manuscript. All authors read and approved the final version of the manuscript.

## Funding

## Competing interests

The authors declare no competing interests.
