## [Peer Review File · Communications Biology]

Reviewers' comments:

Reviewer #1 (Remarks to the Author):

Previously, Kurzchalia group has made an extensive work on the role of cholesterol in reproductive development and suppression of dauer formation in *C. elegans*. In this study Schemisser et al. follow up on these studies identifying sterol binding proteins involved in the mobilization of cholesterol from internal stores during the transition from dauer arrest to grow in *C. elegans*. They further present evidence suggesting that cholesterol activates mTORC1 at the lysosomes to boost growth during dauer exit.

In my opinion, the paper identifies new players of "cholesterol mobilization" in *C. elegans* and sets the ground for better understanding the role of cholesterol in dauer exit. The data presented is not enough to provide a clear mechanistic understanding of cholesterol activation of mTORC1 in *C. elegans*, but it is a step in the right direction and has the potential to promote further discoveries in the role of sterols in TOR signaling .

SCL-12 and SCL-13 are very similar but the authors did not determine whether SCL-13 binds Top Fluor cholesterol. Did the authors tested whether single mutants of either SCL12 and SCL13 compromise dauer exit? This experiment would be useful to determine whether the SCL proteins have redundant functions. This will strength the conclusion that both proteins are necessities to sequester cholesterol in the gut lumen of dauers (line 160).

The term dauer exit and growth is confusing along the manuscript. For example, in line 230 the authors say that in Fig. 5A they analyze dauer exit in *daf-2* worms fed with RNAi against *rab-5* and *rab-7*. However, in such Fig. they graph growth. I think It will better to illustrate the experiments of Fig. 5A using a graph graphing Exit (% of L4) instead length, in function of time.

It will be interesting to test the localization of Top Fluor cholesterol after treatment of dauers with RNAi against *rab-7*. Would they expect that cholesterol will be located in lipid droplets?

Figs 6A and S3D are redundant. I suggest to eliminate Fig 6A from the main text.

On the basis of oxygen consumption during dauer exit of *daf-2,daf-15-AID* animals treated with auxin the authors conclude that fat deposits were metabolized and spent in vain (NO wain!, line 272). This conclusion should be confirmed measuring fatty acid beta-oxidation or softened.

Fig 6E is quite complicated to follow because contain many curves. This Fig. should be modified in some way to understand the conclusions.

It is not obvious that DA cannot rescue L2* larvae obtained under sterol-depleted conditions (compare bars 1 and 3, Fig. 6H). This conclusion is derived from just two experiments with a high SD. Looking the bar error, it still could be possible that DA partially rescues L2* larvae.

In a previous manuscript, Kurzchalia 's group reported that glycolipids mobilized cholesterol in L2* larvae. Perhaps these findings could be mentioned here to illustrate that a mobilization of cholesterol independent of SCL proteins also takes place in *C. elegans*.

Minor points

Line 15, should be introduced a reference

Line 190, Das instead Das

Reviewer #2 (Remarks to the Author):

The authors found that cholesterol bound Protein SCL12/13 could be sequestered in the gut lumen, and after dauer exit, these proteins are degraded and can release cholesterol for further production of dafachronic acids and activation of mTORC signaling, revealing that the mobilization of sequestered cholesterol stores is the key event for transition from quiescence to growth and cholesterol. The story is relatively complete and interesting, but some questions need to be addressed. So I would like to see a major revision for this manuscript. Here are some comments:

1. Some proteins bind cholesterol in mammals, but its homologs can only bind some sterols in lower species. Given to that possibility, can you verify that the SCL-12/13 really bind cholesterol in worms? The CBM motif can bind sterols as well. For the binding affinity assay, can you make a standard curve and show the K_d value? Please make the curves for the competition binding assay as well.
2. For Figure 2D, can you add a time point at 12h to make it mainly consistent with Figure 2B?
3. For verifying that the endosomal pathway could be responsible for the transport of SCL12/13 or cholesterol to the lysosomes, could you find other candidates in endocytic and lysosome/lysosome-related organelle pathway? Only rab-7 showing strong inhibition is not enough to answer the question. And please add more references for the rab-7 RNAi treatment which disrupt the formation for the intact lysosomes or have a diffusive distribution.
4. Some minor mistakes in the manuscript. Please check more carefully in the paper.
 - 1) Line 57, no reference cited.
 - 2) Line 190, should be DAs.
 - 3) Line 202, -chol/-chol.
 - 4) Line 245, 1 mM versus 13 micro Molar.
 - 5) Line 284, figure 5E, D.
 - 6) Line 432, "nmol", please make it consistent with the whole manuscript.

Reviewer #3 (Remarks to the Author):

Schmeisser and colleagues have discovered that the mobilization of gut cholesterol is essential for *C. elegans* to exit the dauer state, during which the SCL-12 & SCL13 proteins are degraded by lysosome to release the bound cholesterol. Free cholesterol then acts as the precursor for DA and as the activator for mTOR signaling, both are required for the exit. The finding of SCL-12 and SCL13 proteins is novel and important, and may contribute to the understanding of developmental growth in *C. elegans*.

Unfortunately, additional experiments are needed to support the mechanisms proposed by the authors:

1. As for the degradation of SCL-12 by lysosome, the authors observed fluorescent co-localizations for each two of SCL-12, cholesterol and lysosome. Although intact lysosomes are proved to be essential for the exit, functional studies of lysosome digestion may strengthen the conclusion. For instance, when intervenes the degradation process by inhibiting the fusion of endosome with lysosome or by decreasing the acidity of lysosome, will more SCL-12 and cholesterol be retained in the endosome or lysosome?
2. Although the title stated "mobilization of cholesterol induces the transition ... through... mTOR signaling", current data only suggest that both daf-15 and cholesterol are required for mTOR activation or "very likely they act within the same cellular process" as the author stated. More experiments are needed to support that the mTOR signaling are indeed activated by the cholesterol released by SCL-12

and SCL-13. For instance, Western blot examination of mTOR signaling activation, with or without SCL-12 and SCL-13, with or without exogenous HC, etc.

3. When comparing Fig.6F and Fig.3B, I cannot understand why DA supplementation could fully restore recovery rate and growth in *scl-12;daf-2* in Fig.3B, since DA is for the activation of DAF-12 but not for the mTOR signaling that requires cholesterol as shown in Fig.6B.

Rebuttal letter/Response to reviewers

We sincerely thank the reviewers for their valuable suggestions. We appreciated their expertise, insight and common sense in requesting additional experiments. We performed most of the suggested experiments and edits and believe that this made the manuscript more convincing and validated the major scientific points.

Detailed response to reviewers:

Reviewer #1 (Remarks to the Author):

Previously, Kurzchalia group has made an extensive work on the role of cholesterol in reproductive development and suppression of dauer formation in *C. elegans*. In this study Schemeisser et al. follow up on these studies identifying sterol binding proteins involved in the mobilization of cholesterol from internal stores during the transition from dauer arrest to growth in *C. elegans*. They further present evidence suggesting that cholesterol activates mTORC1 at the lysosomes to boost growth during dauer exit.

In my opinion, the paper identifies new players of “cholesterol mobilization” in *C. elegans* and sets the ground for better understanding the role of cholesterol in dauer exit. The data presented is not enough to provide a clear mechanistic understanding of cholesterol activation of mTORC1 in *C. elegans*, but it is a step in the right direction and has the potential to promote further discoveries in the role of sterols in TOR signaling .

SCL-12 and SCL-13 are very similar but the authors did not determine whether SCL-13 binds Top Fluor cholesterol. Did the authors tested whether single mutants of either SCL12 and SCL13 compromise dauer exit? This experiment would be useful to determine whether the SCL proteins have redundant functions. This will strength the conclusion that both proteins are necessities to sequester cholesterol in the gut lumen of dauers (line 160).

To test whether single mutants can compromise dauer exit, we generated a CRISPR mutant that lacks only *scl-12*. In the *daf-2* background, the strain looks superficially like the parental *daf-2*. *scl-12;daf-2* exhibited a small, yet statistically significant delay in dauer exit that was much less pronounced compared to the double mutant. We added this result to the text and figures (p. 7, fig. S2E). Thus, the deletion of both SCLs is required for influencing dauer exit.

Due to a technical problem, we were not able to repeat the *in vitro* binding assay with purified SCL-13, like we did with SCL-12. Our numerous efforts to obtain enough purified SCL-13 were not successful. Nevertheless, we addressed the reviewer’s comment by performing an *in vivo* binding assay with the *scl-12* mutants, that has a functional *scl-13*: We replaced cholesterol in their food with fluorescent TopFluor cholesterol and let them develop into dauer larvae. Interestingly, we found that the mean fluorescence intensity of cholesterol in the dauer gut of *scl-12* single mutants was almost the same as in the *daf-2* control animals and much higher than in *scl-12&13* double mutants (p. 7, fig. 2F). We expected this because both SCL-12 and SCL-13 contain the same binding sites for cholesterol and we assumed that they can perform each other’s function in the absence of one. We did however see a much greater variability in fluorescence intensity/amount of cholesterol that was stored in the dauer gut when SCL-12 was missing, as indicated by a high standard deviation. Some worms that lack SCL-12 could not store as much cholesterol, and some could store it as well as the *daf-2* strain containing both forms. We conclude that SCL-13 does indeed bind cholesterol like SCL-12, but that in some individuals a higher binding capacity is needed, so we assume it is an effect of quantity rather than quality.

The term dauer exit and growth is confusing along the manuscript. For example, in line 230 the authors say that in Fig. 5A they analyze dauer exit in *daf-2* worms fed with RNAi against

rab-5 and *rab-7*. However, in such Fig. they graph growth. I think It will better to illustrate the experiments of Fig. 5A using a graph graphing Exit (% of L4) instead length, in function of time.

We agree. We re-evaluated the raw data of fig. 5A (microscopic images) and plotted Exit (% of L4) 48h after induction of dauer exit (fig. S3A). Additionally, we unified the nomenclature of dauer exit and growth/length throughout the manuscript. *rab-5* RNAi could not be investigated because worms died before reaching the dauer state (it was not shown in any figure before, just an explanation in the text of the manuscript).

It will be interesting to test the localization of Top Fluor cholesterol after treatment of dauers with RNAi against *rab-7*. Would they expect that cholesterol will be located in lipid droplets?

Indeed, we found that in dauers that were treated with RNAi against *rab-7* and TopFluor cholesterol, we found cholesterol less accumulated in the gut and more in the periphery, most likely in lipid droplets. We added a graph and representative images to the figures (p. 9, fig. S3B and S3C).

Figs 6A and S3D are redundant. I suggest to eliminate Fig 6A from the main text.

Fig. S3D is now fig. S4D. We would like to point out that, although the depicted experiments are similar, they are not identical. We performed the experiment in different genetic backgrounds, in *daf-2* (fig. 6A) and in wt (starvation dauers; fig. S4D), to check whether the genetic background has an effect on the main outcome. Our observations suggest that *daf-15* is necessary for the growth activation upon dauer exit, regardless on the way of induction of the dauer state.

On the basis of oxygen consumption during dauer exit of *daf-2,daf-15-AID* animals treated with auxin the authors conclude that fat deposits were metabolized and spent in vain (NO wain!, line 272). This conclusion should be confirmed measuring fatty acid beta-oxidation or softened.

We tested our hypothesis by staining the fat deposits of worms with the lipophilic dye Nile Red. We found a tendency towards smaller fat deposits in *daf-15::mNG::AID;TIR1;daf-2* treated with auxin at 4 days after induction of exit, and a significant decrease after 10 days, when they are in the midst of dying (p. 10, fig. S4F). This strengthens our assumption that fat deposits are spent in vain as DAF-15-dependent growth and mitochondrial activation/metabolic switch toward OxPhos do not seem to be coupled.

Fig 6E is quite complicated to follow because contain many curves. This Fig. should be modified in some way to understand the conclusions.

We agree, the figure was overloaded. To make it easier to understand, we removed the EV conditions from fig. 6E and only show *rab-7* RNAi treatment. We added the full figure, including EV, to the supplement (fig. S4G).

It is not obvious that DA cannot rescue L2* larvae obtained under sterol-depleted conditions (compare bars 1 and 3, Fig. 6H). This conclusion is derived from just two experiments with a high SD. Looking the bar error, it still could be possible that DA partially rescues L2* larvae.

We repeated the experiment one more time and found in 3 independent experiments that 75%, 93%, and 96% of all L2* larvae obtained under sterol-depleted conditions remain arrested when treated with DA. An unpaired t-test comparing Ctrl and DA treatment showed $p = 0.1497$, and comparing cholesterol and DA $p = 0.0002$. We updated the figure and included these statistics.

In a previous manuscript, Kurzchalia's group reported that glycolipids mobilized cholesterol in L2* larvae. Perhaps these findings could be mentioned here to illustrate that a mobilization of cholesterol independent of SCL proteins also takes place in *C. elegans*.

We added Boland et al, 2017 to the discussion (p. 14) and updated it in the reference list.

Minor points

Line 15, should be introduced a reference

Line 190, DAs instead Das

We corrected these.

Reviewer #2 (Remarks to the Author):

The authors found that cholesterol bound Protein SCL12/13 could be sequestered in the gut lumen, and after dauer exit, these proteins are degraded and can release cholesterol for further production of dafachronic acids and activation of mTORC signaling, revealing that the mobilization of sequestered cholesterol stores is the key event for transition from quiescence to growth and cholesterol. The story is relatively complete and interesting, but some questions need to be addressed. So I would like to see a major revision for this manuscript. Here are some comments:

1. Some proteins bind cholesterol in mammals, but its homologs can only bind some sterols in lower species. Given to that possibility, can you verify that the SCL-12/13 really bind cholesterol in worms? The CBM motif can bind sterols as well. For the binding affinity assay, can you make a standard curve and show the Kd value? Please make the curves for the competition binding assay as well.

Thank you for these suggestions. Due to a technical problem, we were not able to repeat the *in vitro* binding assay with purified SCL-13, like we did with SCL-12. Our numerous efforts to obtain enough purified SCL-13 were not successful. Instead, we performed an *in vivo* binding assay with *scl-12* single mutants that has a functional *scl-13*, to complement *in vivo* data from *scl-12&13* double mutants. We replaced cholesterol in their food with fluorescent TopFluor cholesterol and let them develop into dauer larvae. Expectedly, we found that the mean fluorescence intensity of cholesterol in the dauer gut of *scl-12* single mutants lay between *daf-2* control animals and *scl-12&13* double mutants, which showed less than 30% of the TopFluor cholesterol fluorescence of wt (fig. 2F). We did however see a much greater variability in fluorescence intensity/amount of cholesterol that was stored in the dauer gut when SCL-12 was missing, as indicated by a high standard deviation. Some worms that lack SCL-12 could not store as much cholesterol, and some could store it as well as the *daf-2* strain containing both forms. We conclude that SCL-13 does indeed bind cholesterol like SCL-12, but that in some individuals a higher binding capacity is needed, so we assume it is an effect of quantity rather than quality. The *scl-12* single mutant looks superficially like the parental *daf-2*. They exhibited a small, yet statistically significant delay in dauer exit that was much less pronounced compared to the double mutant. We added this result to the text and figures (p. 7, fig. S2E). Thus, the deletion of both SCLs is required for influencing dauer exit.

2. For Figure 2D, can you add a time point at 12h to make it mainly consistent with Figure 2B?

We added a 12h time point to fig. 2D.

3. For verifying that the endosomal pathway could be responsible for the transport of SCL12/13 or cholesterol to the lysosomes, could you find other candidates in endocytic and lysosome/lysosome-related organelle pathway? Only rab-7 showing strong inhibition is not

enough to answer the question. And please add more references for the *rab-7* RNAi treatment which disrupt the formation for the intact lysosomes or have a diffusive distribution.

Initially, we investigated many players of the apical trafficking, ESCRT and endocytic/lysosome-related organelle pathways, such as RAB-11, PEPT-1, SEC-15, CAV-2, GLO-3, GLO-1, GLO-4, RME-1, RAB-5, VPS-32.2, and PTC-3. RNAis against these were either lethal before worms reached dauer, or did not show any dauer exit phenotype. We updated this information in the text of the manuscript. Since only *rab-7* showed a pronounced phenotype, we came to our hypothesis that the formation of functional lysosomes is an essential process for dauer exit. To dig deeper in that direction and to strengthen our hypothesis, we tried to selectively disrupt lysosomes with the drug glycyl-L-phenylalanine 2-naphthylamide (GPN) that is supposed to do so in tissue culture, however this did not lead to any noticeable effects in *C. elegans*. We did find an effect on lysosomal morphology with RNAis against *hh-30* and *Imp-1*, and these treatments also show a small delay in dauer exit, however not as pronounced as *rab-7* (lysosomes seem to be less vulnerable to disruptions in worms). We added this to the text and figures (p. 9, fig. S3D and S3E). Regarding references for the *rab-7* RNAi treatment, it has to our knowledge not been shown before that *rab-7* RNAi treatment disrupts lysosomes in dauers. We clarified in the text that this is our result, which was not clear before.

4. Some minor mistakes in the manuscript. Please check more carefully in the paper.

- 1) Line 57, no reference cited.
- 2) Line 190, should be DAs.
- 3) Line 202, -chol/-chol.
- 4) Line 245, 1 mM versus 13 micro Molar.
- 5) Line 284, figure 5E, D.
- 6) Line 432, "nmol", please make it consistent with the whole manuscript.

We corrected these.

Reviewer #3 (Remarks to the Author):

Schmeisser and colleagues have discovered that the mobilization of gut cholesterol is essential for *C. elegans* to exit the dauer state, during which the SCL-12 & SCL13 proteins are degraded by lysosome to release the bound cholesterol. Free cholesterol then acts as the precursor for DA and as the activator for mTOR signaling, both are required for the exit. The finding of SCL-12 and SCL13 proteins is novel and important, and may contribute to the understanding of developmental growth in *C. elegans*.

Unfortunately, additional experiments are needed to support the mechanisms proposed by the authors:

1. As for the degradation of SCL-12 by lysosome, the authors observed fluorescent co-localizations for each two of SCL-12, cholesterol and lysosome. Although intact lysosomes are proved to be essential for the exit, functional studies of lysosome digestion may strengthen the conclusion. For instance, when intervenes the degradation process by inhibiting the fusion of endosome with lysosome or by decreasing the acidity of lysosome, will more SCL-12 and cholesterol be retained in the endosome or lysosome?

Thank you for this suggestion. Initially, we investigated many players of the apical trafficking, ESCRT and endocytic/lysosome-related organelle pathways, such as RAB-11, PEPT-1, SEC-15, CAV-2, GLO-3, GLO-1, GLO-4, RME-1, RAB-5, VPS-32.2, and PTC-3. RNAis against these were either lethal before worms reached dauer, or did not show any dauer exit phenotype. We updated this information in the text of the manuscript. Since only *rab-7* showed a pronounced phenotype, we came to our hypothesis that the formation of functional lysosomes is an essential process for dauer exit. To dig deeper in that direction and to

strengthen our hypothesis, we tried to selectively disrupt lysosomes with the drug glycy-L-phenylalanine 2-naphthylamide (GPN) that is supposed to do so in tissue culture, however this did not lead to any noticeable effects in *C. elegans*. We did find an effect on lysosomal morphology with RNAis against *hlh-30* and *Imp-1*, and these treatments also show a small delay in dauer exit, however not as pronounced as *rab-7* (lysosomes seem to be less vulnerable in worms). We added this to the text and figures (p. 9, fig. S3D and S3E). We also performed a TopFluor cholesterol staining in dauers that had been treated with *rab-7* RNAi. We found that cholesterol is less accumulated in the dauer gut and more in the periphery, most likely in lipid droplets. We added a graph and representative images to the figures (p. 9, fig. S3B and S3C) and updated the text.

2. Although the title stated “mobilization of cholesterol induces the transition ... through... mTOR signaling”, current data only suggest that both *daf-15* and cholesterol are required for mTOR activation or “very likely they act within the same cellular process” as the author stated. More experiments are needed to support that the mTOR signaling are indeed activated by the cholesterol released by SCL-12 and SCL-13. For instance, Western blot examination of mTOR signaling activation, with or without SCL-12 and SCL-13, with or without exogenous HC, etc.

We agree that the link between mTOR, SCL-bound cholesterol, and dauer exit had to be strengthened. We thought that this would be best achieved by a downstream hyperactivation of mTOR: If *scl-12&13* knockout leads to impaired dauer exit via a pathway that inhibits mTOR, a downstream of SCL-12 and SCL-13 activation of mTOR should be able to rescue the dauer exit phenotype. For this, we used an RNAi against the gene coding for Nitrogen Permease Regulator Like 3 (*npri-3*), which has been shown to activate mTORC1 constitutively in *C. elegans* {Zhu, 2013 #24241}. The NPRL-2/NPRL-3 complex, similar to its mammalian orthologue GATOR1, represses mTORC1 and loss of either NPRLs causes a robust mTORC1 hyperactivation {Bar-Peled, 2013 #25265}. Indeed, 100% of the *scl-12&13;daf-2* animals grown on *npri-3* RNAi could exit the dauer state normally, confirming that SCL-12/13-mediated cholesterol transport and mTOR activation are indeed functionally linked. We added this important experiment to the manuscript and figures (p. 11, fig. 6I).

3. When comparing Fig.6F and Fig.3B, I cannot understand why DA supplementation could fully restore recovery rate and growth in *scl-12&13;daf-2* in Fig.3B, since DA is for the activation of DAF-12 but not for the mTOR signaling that requires cholesterol as shown in Fig.6B.

We suggest an explanation to the at the first glance puzzling results. Both DA production and mTOR activation are needed to exit the dauer state – one induces a transcriptional switch, the other mediates growth. Both pathways are dependent on cholesterol that is stored in the dauer state via SCL-12 and SCL-13. We showed that the growth deficit of animals treated with *rab-7* RNAi during dauer exit, i.e. without functional lysosomes, could be rescued with high doses of cholesterol, but not DA (fig. 5C, D). When DAF-15/mTOR is degraded, however, the effect of high cholesterol doses on *rab-7* RNAi treated dauers is abolished (p. 11, fig. 6E, S4G). Consistently, growth of *daf-15::mNG::AID;TIR1;daf-2* dauers that harbor an additional deletion of *scl-12&13* cannot be rescued by DA when DAF-15 is degraded (p. 11, fig. 6F). Thus, the activity of DAF-15 requires intact lysosomes and the presence of cholesterol, but is not connected to the DA pathway.

We hypothesize that in the presence of exogenous DA, the residual cholesterol observed in the dauer gut in *scl-12&13* double mutants (about 25% of the wt content) and additional cholesterol stored in lipid droplets is sufficient to activate mTOR, because it is not required for the synthesis of DAs. We speculate that there might be other SCLs responsible for the remaining 25% of cholesterol stores in the gut – there are 27 in total encoded in the worm genome and we see an increased expression in dauer compared to L3 in at least two more (SCL-11 and SCL-14).

REVIEWERS' COMMENTS:

Reviewer #1 (Remarks to the Author):

The authors have satisfactorily addressed my concerns. I think the manuscript have to be published

Reviewer #2 (Remarks to the Author):

The author addressed all my questions and I agree to the publish of this paper. Thanks.

Reviewer #3 (Remarks to the Author):

Thanks for the authors responses, they basically provided answers to my previous concerns.